# Warming and redistribution of nitrogen inputs drive an increase in terrestrial nitrous oxide emission factor

E. Harris [1,2,16] ✉, L. Yu [3,4,16], Y-P. Wang [5], J. Mohn [4], S. Henne [4], E. Bai [6], M. Barthel[7], M. Bauters [8], P. Boeckx [8], C. Dorich[9], M. Farrell [10], P. B. Krummel [5], Z. M. Loh [5], M. Reichstein [11], J. Six [7], M. Steinbacher [4], N. S. Wells [12,13], M. Bahn [2] & P. Rayner [14,15]

Anthropogenic nitrogen inputs cause major negative environmental impacts, including emissions of the important greenhouse gas $N_2O$. Despite their importance, shifts in terrestrial N loss pathways driven by global change are highly uncertain. Here we present a coupled soil-atmosphere isotope model (IsoTONE) to quantify terrestrial N losses and $N_2O$ emission factors from 1850-2020. We find that N inputs from atmospheric deposition caused 51% of anthropogenic $N_2O$ emissions from soils in 2020. The mean effective global emission factor for $N_2O$ was $4.3 \pm 0.3\%$ in 2020 (weighted by N inputs), much higher than the surface area-weighted mean ($1.1 \pm 0.1\%$). Climate change and spatial redistribution of fertilisation N inputs have driven an increase in global emission factor over the past century, which accounts for 18% of the anthropogenic soil flux in 2020. Predicted increases in fertilisation in emerging economies will accelerate $N_2O$-driven climate warming in coming decades, unless targeted mitigation measures are introduced.

Nitrous oxide ($N_2O$) is a long-lived greenhouse gas and a key stratospheric ozone-depleting substance[1,2]. The atmospheric $N_2O$ mole fraction has increased from ~270 nmol mol$^{-1}$ in the preindustrial era to >332 nmol mol$^{-1}$ today[2,3]. The primary global source of $N_2O$ is production during N cycling by microbiota in soils . Soil N cycling also releases NO and $N_2$, which directly impact tropospheric ozone

production, climate, and soil N content and loss pathways[4,5]. N cycling and thus N gas production are strongly enhanced by direct and indirect anthropogenic N inputs[2,6–8]. Agricultural fertilisation accounts for around two thirds of anthropogenic N inputs[9], with the remainder contributed by biological $N_2$ fixation and deposition of $NO_x$ and $NH_3$. Most anthropogenic N is not incorporated into crops or soils but lost

[1]Swiss Data Science Centre, ETH Zurich, 8092 Zurich, Switzerland. [2]Functional Ecology Research Group, Institute of Ecology, University of Innsbruck, 6020 Innsbruck, Austria. [3]Institute of Environment and Ecology, Tsinghua Shenzhen International Graduate School (SIGS), Tsinghua University, Shenzhen 518055, China. [4]Laboratory for Air Pollution & Environmental Technology, Empa, Swiss Federal Laboratories for Materials Science and Technology, 8600 Duebendorf, Switzerland. [5]Climate Science Centre, CSIRO Oceans and Atmosphere, Aspendale, VIC 3195, Australia. [6]Key Laboratory of Geographical Processes and Ecological Security of Changbai Mountains, Ministry of Education, School of Geographical Sciences, Northeast Normal University, Changchun 130024, China. [7]Department of Environmental Systems Science, ETH Zurich, 8092 Zurich, Switzerland. [8]Isotope Bioscience Laboratory - ISOFYS, Department of Green Chemistry and Technology, Ghent University, Coupure Links 653, 9000 Ghent, Belgium. [9]Natural Resource Ecology Laboratory, Colorado State University, Fort Collins 80523 CO, USA. [10]CSIRO Agriculture and Food, Locked bag 2, Glen Osmond, SA 5064, Australia. [11]Department of Biogeochemical Integration, Max Planck Institute for Biogeochemistry, Jena, Germany. [12]Centre for Coastal Biogeochemistry, Southern Cross University, Lismore, NSW 2480, Australia. [13]Department of Soil and Physical Sciences, Agriculture and Life Sciences, Lincoln University, Lincoln 7647, New Zealand. [14]School of Geography, Earth and Atmospheric Sciences, University of Melbourne, Parkville, VIC 3052, Australia. [15]Melbourne Climate Futures Climate and Energy College, University of Melbourne, Parkville, VIC 3052, Australia. [16]These authors contributed equally: E. Harris, L. Yu. ✉e-mail: eliza.harris@sdsc.ethz.ch

to the environment[10], representing huge monetary losses for the agricultural sector and causing a cascade of environmental problems[11–13]. In the coming decades, N inputs are expected to grow in line with increasing agricultural production, and to shift towards tropical regions and emerging economies as strict N pollution controls are introduced in many developed regions[14–18]. Effective management of N fertiliser to achieve high N use efficiency is key to balancing food production with environmental protection and thus reducing $N_2O$ emissions[19,20]. However, mitigation is challenging, and $N_2O$ emissions are currently exceeding the highest predicted scenarios[2].

N is lost from terrestrial ecosystems through several major pathways: Microbial and abiotic *N gas production* in soils; runoff and *leaching* of N species; and ammonia *volatilization*. Despite their importance, expected changes in N losses in the coming century are not well known[11,21]. Nitrification (aerobic) and denitrification (anaerobic) are the main processes emitting N gases (NO, $N_2O$ and $N_2$) from soils. The proportion of N inputs released as particular N gases can be described with the emission factor (EF); for example, an EF for $N_2O$ of 2% means that 2% of annual N inputs are released as $N_2O$. On the global scale, the impacts of climate change on N-gas production processes are poorly known: Warming is generally expected to enhance microbial activity and increase N-gas EFs, however interactions between factors such as N availability, plant growth, and precipitation changes are poorly constrained. Moreover, it is unknown if increased nitrification or denitrification rates would in fact lead to increased N gas production[22–24]. The proportion of N lost by leaching globally is not expected to change significantly over time in response to warming due to the contrasting effects of increased N mineralization and reduced moisture availability[2,25–27]. However, leaching losses and predicted responses to warming vary widely between different regions depending on soil, hydrological and ecosystem parameters, and may also be strongly affected by precipitation regime changes[28,29]. Moreoever, increasing atmospheric $CO_2$ generally enhances plant growth and N uptake, thus impacting the availability of N in soils for different loss pathways[30,31]. Process models have been used to simulate N loss pathways in a changing climate (e.g.[32–34]). These models used generally require large amounts of input data and parameterisations as well as high computing power, which makes it difficult to iteratively constrain and optimize model parameters with observations using typical inversion frameworks and likelihood approaches, and complicates investigations of global or long-term emissions. Top-down modelling efforts can give robust estimates of global emissions for recent decades[35–37], however without the incorporation of isotopes, these approaches cannot provide mechanistic information.

The isotopic composition of soil N ($\delta^{15}N_{soil}$) has been proposed as an integrated indicator of N loss partitioning in natural systems[38–41]. Leaching of soluble N species (eg. $NO_3^-$) involves very low isotopic fractionation, while losses through ammonia volatilization and N gas production strongly favour $^{14}N$ and thus cause $^{15}N$ enrichment in the remaining soil N pool[39]. Observations have shown that mean global $\delta^{15}N_{soil}$ is elevated relative to N inputs, reflecting significant production of gaseous N species[38,39], although the relationship may be unpredictable in some regions due to N immobilization[41]. Previous studies have used $\delta^{15}N_{soil}$ models to constrain N losses at individual sites or globally for natural ecosystems[38,39], however this approach has not been applied to estimate temporal changes in N loss pathways. Furthermore, the availability of complementary datasets to validate previous $\delta^{15}N_{soil}$ models has been limited to short-term measurements from individual sites. This approach can be extended to include records of atmospheric $N_2O$ isotopic composition, which reflects $N_2O$ production pathways[42–44], and can be used to validate results from studies of $\delta^{15}N_{soil}$. Atmospheric $N_2O$ isotopic composition can be described with the $N_2O$ bulk 15-N isotopic composition ($\delta^{15}N^{bulk}$), hereafter abbreviated as $\delta^{15}N$, and tje $N_2O$ 15-N isotopic site preference, hereafter referred to as $\delta^{15}N^{SP}$ (see ref. 45 for definitions and review).

Recent advances in spectroscopic isotope instrumentation have delivered high precision long-term time series of background tropospheric $N_2O$ mixing ratio and isotopic composition[43,45], allowing an integrated view of $N_2O$ sources and sinks. These results have been used to estimate total anthropogenic $N_2O$ emissions based on two-box models of the atmosphere[42,45,46], however this approach cannot give spatially resolved information on $N_2O$ sources.

Here we aim to gain new insight into the N loss processes underlying global $N_2O$ emissions and their spatiotemporal patterns, in order to understand how N losses are changing under the influence of anthropogenic activities and climate change. We use an artificial neural network to estimate a global isoscape of natural soil $\delta^{15}N$. This is used to initialise a soil module to simulate spatially resolved terrestrial N losses via leaching, volatilization and gas production pathways. $N_2O$ emissions from the soil module are released to a two-box atmospheric module to simulate $N_2O$ mixing ratio and isotopic composition from the preindustrial era to the present day. The coupled model framework, 'IsoTONE', is optimized within a Bayesian framework using a high precision time series of $N_2O$ mixing ratio and isotopic composition from several background sites, as well as estimates of $N_2O$ emission factors from the Global $N_2O$ Database.

## Results and discussion
### Terrestrial $N_2O$ emissions

12 key parameters in the IsoTONE model were optimized using a Markov Chain Monte Carlo (MCMC) approach with 120 000 iterations, described in detail in Supplementary Note 3 and Supplementary Table 1. Total terrestrial $N_2O$ emissions from the optimised model (Table 1) agree well with previous results, providing confidence in the isotopic basis of the model, for example: Total terrestrial $N_2O$ emissions for 1860, 2010 and 2020 were $5.3 \pm 0.4$, $12.6 \pm 1.2$ and $13.9 \pm 1.4$ Tg $N_2O$-N $a^{-1}$ respectively (2020 results shown in Supplementary Fig. 12), showing good agreement with 1860 and 2010 estimates of $6.3 \pm 1.1$ and $10 \pm 2.2$ Tg $N_2O$-N $a^{-1}$ from the $N_2O$ Model Intercomparison Project[8]. The range of 12.6 to 13.9 Tg $N_2O$-N $a^{-1}$ modelled for 2007-2016 additionally agrees with a recent global meta-analysis estimating total terrestrial emissions of 12.2-23.4 Tg $N_2O$-N $a^{-1}$ for the same period[2], and with the mean of 12.9 Tg $N_2O$-N $a^{-1}$ from a meta-analysis by Scheer et al.[47]. Total global soil NO emissions for 2010 were estimated to be $13.7 \pm 3.9$ Tg NO-N $a^{-1}$, agreeing well with the high end of estimates from a meta-analysis suggesting global soil NO emissions of 1.8-12.3 Tg NO-N $a^{-1}$[48].

Natural $N_2O$ emissions are dominated by N inputs from fixation ($4.7 \pm 0.8$ Tg $N_2O$-N $a^{-1}$, 90% of natural emissions). In contrast, anthropogenic soil emissions are dominated by N inputs from deposition (70% of 1940 and 51% of 2020 emissions), although fertilisation is becoming increasingly important (14% of 1940 and 22% of 2020 emissions) (Table 1). The spatial distribution of emissions from fertilisation, deposition and fixation for natural and anthropogenic soils for 2020 is shown in Supplementary Fig. 12, with 'anthropogenic emissions' referring to all emissions above the preindustrial baseline, thus accounting for both direct and indirect anthropogenic $N_2O$ sources. The spatial distribution of $N_2O$ emissions agrees well with inversion estimates from the Copernicus Atmospheric Monitoring Service (CAMS[49];) using data from 123 ground-based sites (Fig. 1), with significant differences only seen in small isolated regions. The IsoTONE framework assumes that land use changes—aside from fertiliser use and other N inputs, which are explicitly provided to the model—have had a minor impact on $N_2O$ emission factors at an annual timescale, compared to the major impact of pre-existing variability in emission factors driven by climate and soil parameters. This assumption is supported by the good agreement between CAMS inversion results and IsoTONE emission estimates. The largest differences are seen in tropical South America, Africa and Australia, which may be due to the scarcity of atmospheric monitoring stations available to constrain

**Table 1 | Changing characteristics of the total and anthropogenic terrestrial N$_2$O flux at the beginning of the anthropocene (1850) and through the past century**

| | | 1850 | 1940 | 1980 | 2020 |
|---|---|---|---|---|---|
| Total soil flux | Tg N$_2$O-N a$^{-1}$ | 5.3 ± 0.4 | 6.3 ± 0.5 | 8.7 ± 0.6 | 12.4 ± 0.8 |
| Natural soil flux | Tg N$_2$O-N a$^{-1}$ | 5.3 ± 0.4 | | | |
| Anthropogenic soil emissions | Tg N$_2$O-N a$^{-1}$ | 0 | 1.0 ± 0.7 | 3.4 ± 0.7 | 7.1 ± 0.9 |
| Growth rate of emissions | Gg N$_2$O-N a$^{-1}$ a$^{-1}$ | 19 ± 10 | 23 ± 0.1 | 60 ± 5 | 134 ± 6 |
| Nat. soil emissions—deposition N | Tg N$_2$O-N a$^{-1}$ | 0.5 ± 0.1 | | | |
| Nat. soil emissions—fixation N | Tg N$_2$O-N a$^{-1}$ | 4.7 ± 0.5 | | | |
| Anth. soil emissions—fertilisation N | Tg N$_2$O-N a$^{-1}$ | 0 | 0.1 ± 0.1 | 0.9 ± 0.2 | 1.7 ± 0.4 |
| Anth. soil emissions—deposition N | Tg N$_2$O-N a$^{-1}$ | 0 | 0.7 ± 0.1 | 1.8 ± 0.2 | 3.6 ± 0.3 |
| Anth. soil emissions—fixation N | Tg N$_2$O-N a$^{-1}$ | 0 | 0.2 ± 0.6 | 0.7 ± 0.7 | 1.8 ± 0.6 |
| Anth. emissions not from soils | Tg N$_2$O-N a$^{-1}$ | 0 | 1.0 ± 0.7 | 1.4 ± 0.7 | 1.7 ± 0.9 |
| $\delta^{15}$N, natural soil emissions | ‰ | −22.4 ± 2.7 | | | |
| $\delta^{15}$N, anthrop. soil emissions | ‰ | NA | −15.3 ± 1.1 | −17.0 ± 1.1 | −18.7 ± 1.1 |
| $\delta^{15}$N$^{SP}$, natural soil emissions | ‰ | 6.7 ± 0.6 | | | |
| $\delta^{15}$N$^{SP}$, anthrop. soil emissions | ‰ | NA | 5.9 ± 0.2 | 6.0 ± 0.2 | 6.8 ± 0.2 |
| EF, area-weighted | % | 1.1 ± 0.1 | | | |
| EF, N input-weighted | % | 3.6 ± 0.7 | 3.8 ± 0.7 | 4.0 ± 0.8 | 4.3 ± 0.3 |

The natural flux and the area-weighted emission factors (EFs) do not vary temporally. Growth rate is the 10-year mean growth rate centred on the year of interest, ie. the 10-year average for 2000 is the average from 1995-2005. The area-weighted EF is the mean global EF calculated using the areas of grid cells as weights; the N-input weighted EF is calculated using the total N inputs of grid cells as weights.

inversion estimates in these regions[49]. Differences may also relate to extensive land use change and cultivation of N-fixing soybean crops in South America. Furthermore, fundamental differences in N cycling and immobilization in tropical and Arctic regions may affect the accuracy of the IsoTONE model in these regions, as most experimental and field studies were conducted in temperate soils[40,50]. Total N$_2$O emissions agree very well between the two models, with 13.3 ± 0.1 Tg N$_2$O-N a$^{-1}$ and 13.9 ± 0.9 Tg N$_2$O-N a$^{-1}$ predicted for 2019 by the CAMS inversion and IsoTONE respectively.

The incorporation of isotopic composition means that, unlike previous global models, the IsoTONE framework distinguishes between different N$_2$O production pathways (Supplementary Note 4). The model results furthermore suggest that on average, laboratory-measured fractionation factors for these pathways are only expressed at ~55% on average in soils (frac_ex = 0.55 ± 0.05, Supplementary Table 1). This is a global average which we expect to vary widely between individual sites depending on soil structure, moisture, and the depth of N$_2$O production and consumption. For example, soils with small N cycling and gas diffusion rates would be expected to have higher isotopic fractionation during N loss processes compared to soils with high diffusivity. In 2020, 60% of N$_2$O emitted from soils (7.5 ± 0.4 Tg N$_2$O-N a$^{-1}$) was produced from denitrification and 39% (4.7 ± 0.8 Tg N$_2$O-N a$^{-1}$) from nitrification. The contribution of nitrification to soil N$_2$O emissions has decreased very slightly from 40% in 1850 to 39% in 2020 (Supplementary Fig. 10). The spatial distribution of nitrification-N$_2$O from IsoTONE agrees very well with the global map produced by Pan et al.[51], with high nitrification in areas such as the Amazon basin, sub-Saharan Africa, Europe and coastal Australia. However the model of ref. 51, centers around total nitrification rate, which is not estimated in IsoTONE, so the results cannot be directly compared.

Globally, just 30% of fertiliser N appears to be available for N cycling (fert_EF_red, Supplementary Table 1)—the remaining fertiliser N is primarily incorporated into harvest, and may also be immobilised via increased soil storage, or lost through leaching pathways which are not explicitly modelled. This is consistent with previous meta-analyses, which show that 15–70% of fertiliser N is taken up by plants, a significant proportion could remain in soils, and 5–25% is unaccounted for in plant or soil pools and thus lost to the pathways modelled in this study[10,52–54]. Therefore, emission factors for fertiliser emissions can be significantly lower than for other N inputs, with implications for the applicability of EF measurements from agricultural sites. Mean global fertiliser N incorporation could be increasing as plant growth and thus N use is enhanced in response to increasing atmospheric CO$_2$. However, water and nutrient limitations will also play a role in regulating plant growth[2], and fertiliser use may moreover increase in response to higher N use[55]. These potential effects are not currently captured in IsoTONE and should be a focus of future model studies.

## The anthropogenic N$_2$O budget: Inputs, losses and trends

The anthropogenic N$_2$O soil flux in 2020 was estimated to be 7.1 ± 0.6 Tg N$_2$O-N a$^{-1}$, close to the highest projected emission scenario (RCP8.5) estimate for 2020[2,56] (Fig. 2 and Supplementary Fig. 11). 1.7 ± 0.4, 3.6 ± 0.3 and 1.8 ± 0.6 Tg N$_2$O-N a$^{-1}$ of the anthropogenic flux were contributed by soil emissions from fertilisation, deposition and fixation N inputs respectively (Table 1), with an additional 1.7 Tg N$_2$O-N a$^{-1}$ from non-soil anthropogenic emission sources (emissions from EDGAR for categories 1A1, 1A3b, 2B and 6, see Methods: Atmospheric N$_2$O module). This agrees very well with a recent ensemble analysis, which estimated a total anthropogenic flux of 7.3 (4.2–11.4) Tg N$_2$O-N a$^{-1}$[12]. We find that deposition N accounts for 41 ± 14% of all anthropogenic emissions in 2020 compared to 19 ± 12% direct emissions from fertilisation, 21 ± 15% from enhanced fixation, and 19% from non-soil sources: Deposition N inputs clearly contribute the majority of anthropogenic emissions. This finding is in contrast to the results of Tian et al.[2], who report that direct N$_2$O emissions from fertilisation are dominant. These contrasting results are due in part to the classification of all emissions above the 1850 baseline (eg. enhanced N$_2$O from natural sites due to warming, fixation and deposition) being classified as anthropogenic in this study, whereas in previous studies some or all of these processes are not considered in the calculation of the anthropogenic burden. Moreover, our results suggest that emission factors could be significantly underestimated from field measurements (see Sections 1 and 1), likely due to the highly dynamic nature of N$_2$O emissions, which are not adequately captured with sparse sampling[20,57,58]. Moreover, measured EFs are often based only on growing season emissions, which can lead to a strong underestimation of annual emissions that could be particularly important in cold

## a) IsoTONE: Total emissions, 2020 (g N$_2$O-N m$^{-2}$ a$^{-1}$)

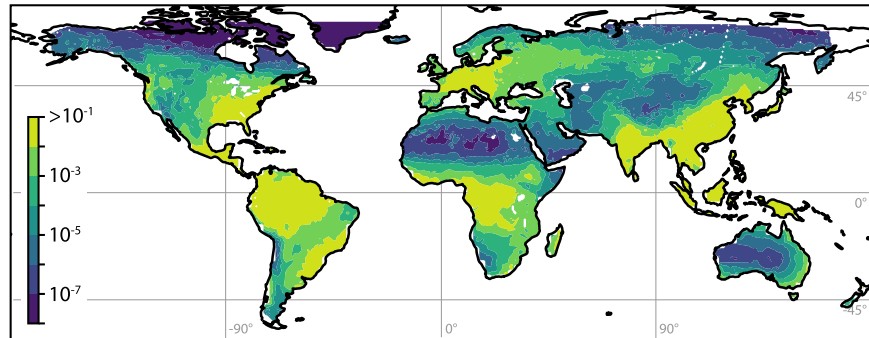

## b) Inversion: Total emissions, 2020 (g N$_2$O-N m$^{-2}$ a$^{-1}$)

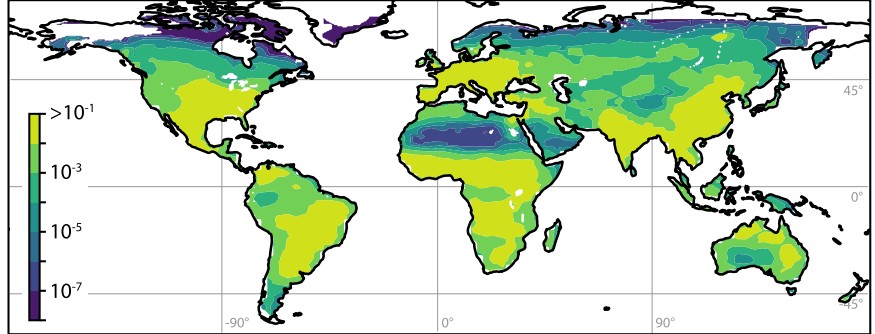

## c) Difference: IsoTONE - inversion (g N$_2$O-N m$^{-2}$ a$^{-1}$)

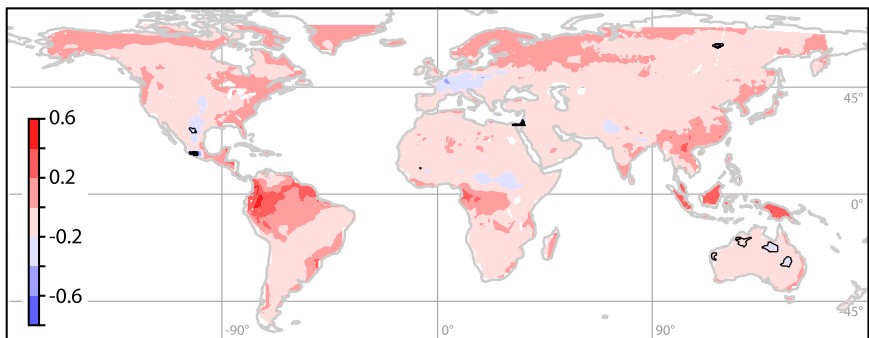

**Fig. 1 | A comparison of total N$_2$O emissions from the IsoTONE and CAMS models for the year 2020.** Modelled total terrestrial N$_2$O fluxes from the IsoTONE model (**a**) and from CAMS (**b**) (R. Thompson, 2021; regridded to 0.5°using linear interpolation) for the year 2019. Both panels use the same logarithmic colour scale. The difference between the IsoTONE and inversion estimates is shown in **c**; areas where the difference is significant compared to the uncertainty are highlighted with black outlines. Maps generated with Cartopy (Met Office, 2015,[132]).

regions[59–61]. EFs will be particularly underestimated from field measurements made at agricultural sites, where a significant proportion of N is removed through harvest or immobilization as described by fert_EF_red, thus leading to low EFs. These combined effects will lead to underestimation of deposition emissions when measured EFs are applied at non-agricultural sites in bottom up-N$_2$O emission frameworks.

The 10-year mean growth of total anthropogenic N$_2$O emissions was between 0 and 0.04 Tg N$_2$O-N a$^{-1}$ a$^{-1}$ until ~1945 (Fig. 2). Between 1945 and 1980 the 10-year mean growth rate in emissions increased to 0.12 ± 0.02 Tg N$_2$O-N a$^{-1}$ a$^{-1}$ due to rapid growth in fertiliser inputs, after which it stabilised at around 0.1 Tg N$_2$O-N a$^{-1}$ a$^{-1}$, corresponding to a steady rate of change of around 0.015 nmol mol$^{-1}$ a$^{-1}$ a$^{-1}$ in tropospheric background N$_2$O growth rate. The growth rate of anthropogenic emissions was particularly high between 2010 and 2015 (3-year growth rate up to 0.3 Tg N$_2$O-N a$^{-1}$ a$^{-1}$) compared to the average of the last 50 years (0.09 ± 0.06 Tg N$_2$O-N a$^{-1}$ a$^{-1}$), in agreement with two recent studies[2,37]. This caused the rate of change for the tropospheric N$_2$O burden to peak at 0.026 nmol mol$^{-1}$ a$^{-1}$ a$^{-1}$ (Fig. 2). After 2015, the growth rate of emissions strongly decreased, meaning the 10-year mean growth for 2010–2020 was 0.14 Tg N$_2$O-N a$^{-1}$ a$^{-1}$ and thus within then normal range for the last half-century (Fig. 2). Fluctuations in the growth rate for tropospheric background N$_2$O mixing ratio reflected the growth rate in emissions driven by different input categories. Variability in fixation N inputs dominates subdecadal interannual variability in total terrestrial N$_2$O emissions, such as the 2010–2015 peak, contributing >70% of variability within decadal bins (Fig. 2). Fixation inputs are 2–10 times more variable than other input types at subdecadal timescales, thus accounting for their key role in driving variability in N$_2$O emissions. In contrast, changes in fertilisation inputs drive the changing growth rate of anthropogenic emissions at timescales larger than 1–2 decades. Increases in both fertilisation and deposition are responsible for the strong and constant increase in N$_2$O emissions and tropospheric background mixing ratio over the last

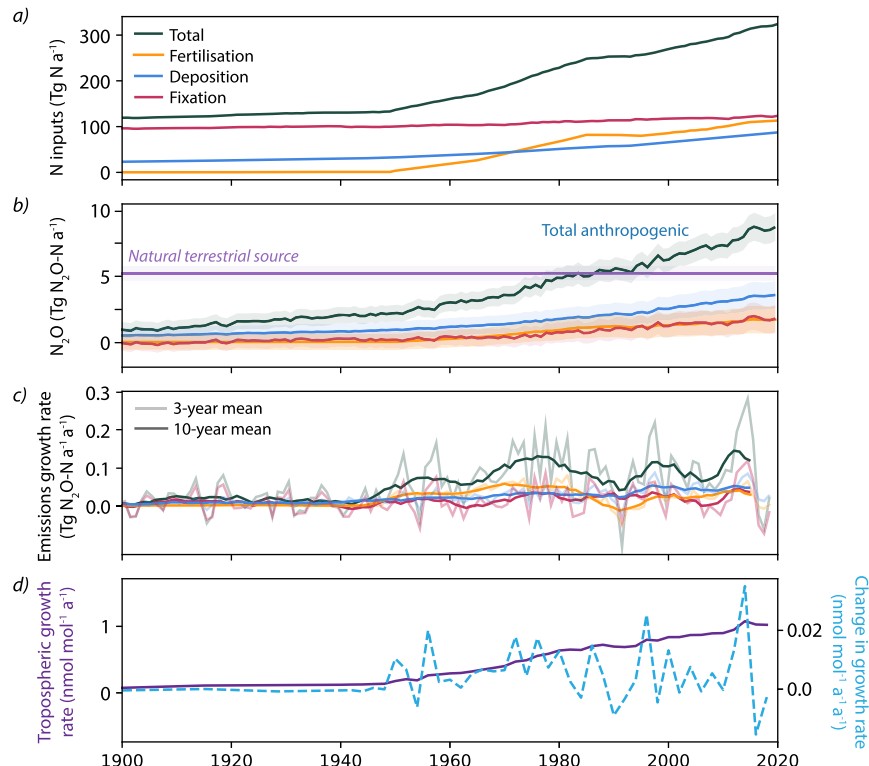

**Fig. 2 | Temporal evolution of N inputs and N₂O emissions, and growth rates of N₂O emissions and N₂O mixing ratio. a** Annual inputs for fertilisation, deposition, fixation, and total N used in the model (see Methods: at the same geographical locations to gapfill the ancillary data for data sources). **b** The anthropogenic flux broken down into N input categories of fertilisation, deposition and fixation, estimated by assuming that all increases in N₂O emissions for all input categories (fixation, fertilisation, deposition) after 1850 are due to anthropogenic influences. Total N₂O emissions, including non-soil N₂O emissions, are also shown. The shaded areas indicate the 1σ uncertainty. N₂O emission data prior to 1900 as well as the breakdown of natural emissions driven by deposition and fixation are shown in Supplementary Fig. 11. **c** The growth rate of N₂O emissions from each input category calculated over 3 and 10 year windows (pale and dark lines respectively, using colours indicated for fertilisation, fixation and deposition). **d** The growth rate in modelled N₂O tropospheric mixing ratio (left axis; purple solid line) as well as the modelled change in mixing ratio growth rate (right axis; blue dotted line).

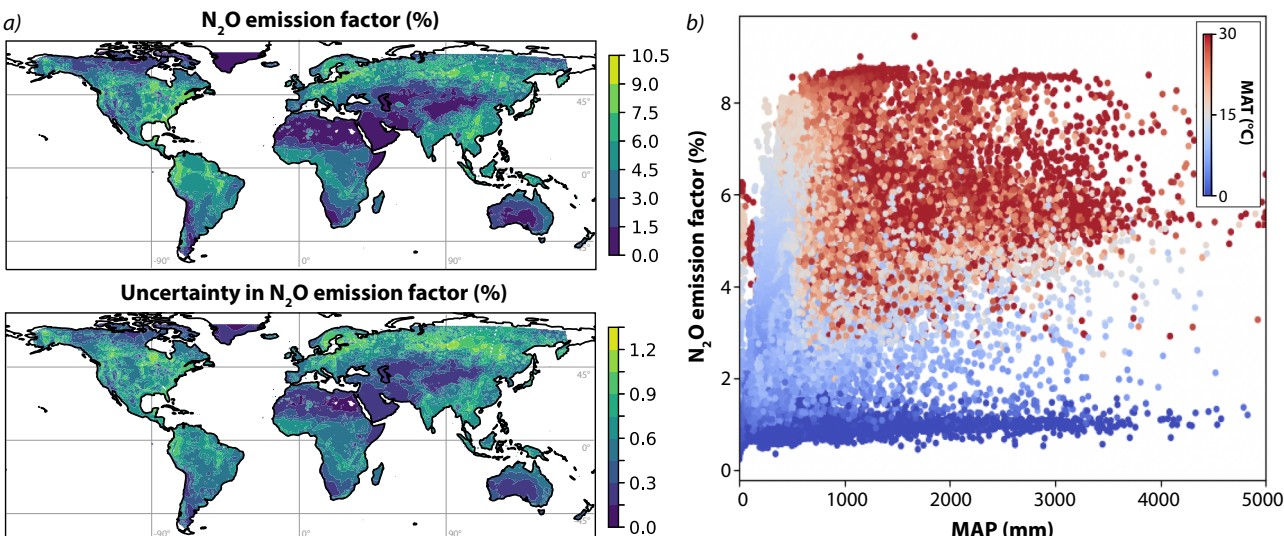

**Fig. 3 | Global map of modelled N₂O emission factors. a** Global gridded N₂O emission factor (upper) and the 1σ uncertainty (lower); **b** Relationship between mean annual precipitation (MAP), mean annual temperature (MAT; point colour) and N₂O emission factor for each grid cell. Maps generated with Cartopy (Met Office, 2015,[132]).

a) **Change in N₂O emissions (1940-2020) only due to changing spatial distribution of fertiliser N inputs**

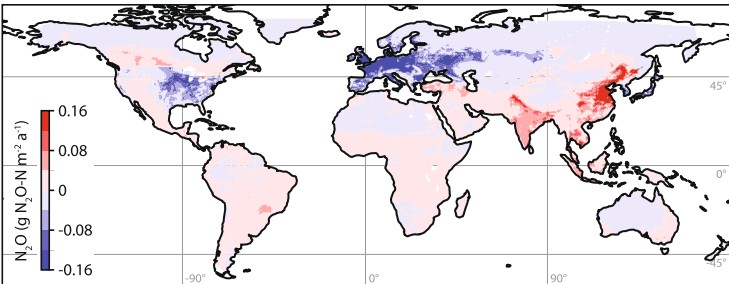

b) **Change in N₂O emissions (1940-2020) only due to temperature dependence of emissions**

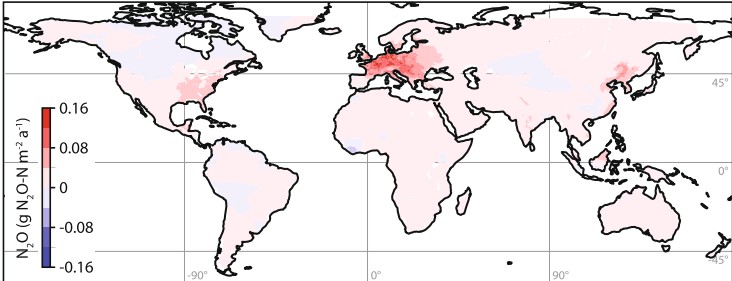

c)

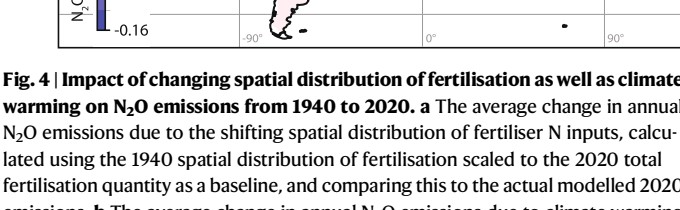

**Fig. 4 | Impact of changing spatial distribution of fertilisation as well as climate warming on N₂O emissions from 1940 to 2020. a** The average change in annual N₂O emissions due to the shifting spatial distribution of fertiliser N inputs, calculated using the 1940 spatial distribution of fertilisation scaled to the 2020 total fertilisation quantity as a baseline, and comparing this to the actual modelled 2020 emissions. **b** The average change in annual N₂O emissions due to climate warming and the temperature dependency of N₂O production, found by calculating 2020 emissions with and without the impact of warming on EFs since 1940. **c** The proportion of anthropogenic N inputs from fertilisation, deposition and fixation in 1940, 1980 and 2020 accounted for within bins defined according to the EF for N₂O. 20 bins were used; 19 were distributed evenly between the parameter minimum and the mean + 3 standard deviations; the highest bin was for all data >mean + 3 standard deviations. Maps generated with Cartopy (Met Office, 2015,[132]).

century. These findings could be impacted by uncertainties in the input datasets, which should be investigated further using targeted isotopic and modelling approaches at the site and regional scale in regions with particularly high uncertainty.

The isotopic composition of the total anthropogenic N₂O source reflects changing emission processes (Supplementary Fig. 11). During the growth rate acceleration of 1945–1980, $\delta^{15}N$ and $\delta^{15}N^{SP}$ of the anthropogenic source were relatively constant, however after 1980 they changed more rapidly. These fluctuations were also observed by Prokopiou et al.[62], who used a two-box model to interpret N₂O source isotopic composition from firn air data, and found similar but more uncertain values for anthropogenic source isotopic composition. $\delta^{15}N$ of the anthropogenic source is higher than the mean soil source (Table 1) and shows a strong decreasing trend, which may indicate an increasing dominance of agricultural emissions with low $\delta^{15}N$-N₂O after 1980[62]. An increasing proportion of agricultural emissions could also explain the trend in anthropogenic source $\delta^{15}N^{SP}$, which approaches the estimated agricultural mean of 7.2 ± 3.8‰[45]. Other explanations for changing source isotopic composition include changes in the extent of pathways such as consumption via N₂O reduction and production via nitrification and denitrification. Variability in $\delta^{15}N^{SP}$ and $\delta^{15}N$ are in opposite directions, and thus unlikely to be caused by changes in the extent of N₂O reduction to N₂, which would increase both the $\delta^{15}N^{SP}$ and $\delta^{15}N$ of remaining N₂O[63]. Furthermore, our results suggest a 1% decrease in nitrification contribution to global N₂O emissions and thus very little change in the nitrification:denitrification ratio, thus this could not account for the observed increase of ~1‰ in $\delta^{15}N^{SP}$ of anthropogenic N₂O based on $\delta^{15}N^{SP}$ endmembers of 0 and 30‰ for N₂O from denitrification and nitrification.

## Drivers of spatiotemporal variability in N₂O emission factors

Soil moisture, N content, mean annual precipitation and soil bulk density (21, 9, 5, 4% of variability respectively) were the main parameters causing broad geoclimatic gradients in N₂O EFs and the proportion of N lost to N gas production between different regions (Fig. 3 and Supplementary Figs. 3 and 13), consistent with previous laboratory and field results[64–67]. Mean annual temperature was important for total N gas production, but not for N₂O EF (5 vs. 0.2% of variability in $f_{gas}$ and EF). Total gas production accounts for the largest proportion of N losses in warm, dry regions, whereas N₂O EFs are highest in non-desert tropical regions, in particular sub-Saharan Africa, southern India, China, and south east Asia, and low in drier and colder areas. This agrees well with results from refs. 2 and 68, as well as in situ measurement studies showing high N losses via both leaching and denitrification from 'leaky' N cycles in tropical regions[69,70]. Geoclimate-driven variability in emission factors can be clearly seen in regions underrepresented in EF compilations, such as Australia, which shows a clear gradient from low to high N₂O EFs between the dry centre and wetter coastal and northern regions, but very little gradient in total N gas production. This pattern, whereby arid regions show relatively high N gas production dominated by NO, but low N₂O EFs, can be seen across central North America, the Sahara, Australia, and central Asia. This suggests global applicability of the experimental results from ref. 71, who used isotopic tracing to show high NO production from arid soils was due to reduced plant N uptake and low leaching.

The spatial distribution of N inputs plays a key role in determining the overall global EF for N₂O (Fig. 4). The mean global EF for N₂O weighted by area was 1.1 ± 0.1%, close to the IPCC default value of 1.4%

for combined direct and indirect emissions[72,73]. However, the mean global EF weighted by total N inputs per grid cell in 2020 was 4.3 ± 0.3%, as a much greater proportion of total N inputs are in 'high emission' temperate and tropical non-arid latitudes compared to dry and cold areas (Fig. 3). This builds upon the observation of Tian et al.[2], who showed that the mean global EF for $N_2O$ from agricultural soils is significantly higher than the IPCC default value. Measurements of mean annual EFs using chambers and similar methodologies are far more time intensive and expensive than measurements of soil $\delta^{15}N$, thus isotopic modelling of the N cycle can be used to understand regional variability in EFs to facilitate upscaling and extrapolation of data from traditional methods. Modelled EFs were compared to the compiled values for croplands used in the meta-analysis of ref. 74 by finding the mean and standard deviation of all EFs reported by Cui et al.[74] in each gridcell. Field EF measurements are laborious and challenging, thus relatively few data points are available—of the 55 478 gridcells with a valid modelled EF, only 179 have one or more EF measurements. The agreement between modelled values and observations was relatively good, with a Spearman correlation coefficient of 0.4 ($p < 0.01$) indicating moderate agreement. However, the slope of 0.15 suggested that measurements may consistently underestimate EFs due to insufficient measurement frequency and duration and the importance of 'hot spots' and 'hot moments' for annual $N_2O$ emission totals[57,75]. The impact of climate change—in particular changing precipitation patterns—on $N_2O$ emissions in rapidly developing regions like sub-Saharan Africa and India is not yet captured by observations, and should be a focus of future studies. Understanding the large-scale impact of moisture availability and other climate parameters on N loss processes through targeted measurements campaigns and model development will be key to predicting interactions between climate change-driven precipitation changes and $N_2O$ emissions.

The proportion of $N_2O$ lost to different pathways shows clear temporal changes, particularly for fertilisation N inputs, which show an increase in mean EF of $N_2O$ from <4% prior to 1940 to >5% in 2020 (Supplementary Fig. 14). This has two major causes: Changing spatial distribution of N inputs and climate warming feedback (Fig. 4). The impact of changing spatial distribution of N inputs on annual $N_2O$ emissions and EFs was estimated by calculating a baseline, using the 1940 spatial distribution of fertilisation scaled to the 2020 total fertilisation quantity, and comparing this to the modelled 2020 maps of emissions and EFs. The shift in fertilisation from North America and other temperate regions (EFs for $N_2O$ of 4–5%) towards emerging economies in warmer regions with higher EFs ($EF_{N_2O}$>7.5%), in particular China, has caused additional emissions of 0.5 ± 0.3 Tg $N_2O$-N $a^{-1}$ since 1940 (Fig. 4c). This effect is only seen for fertilisation N inputs: Deposition and fixation inputs show very minor spatiotemporal changes (Fig. 4c). Deposition inputs are stabilising in China due to vigorous controls on N pollution[76], and changes in $N_2O$ from deposition in coming decades are likely to be of minor importance compared to fertilisation- and climate-driven emissions. The impact of climate warming (Fig. 4b) has led to an increase in total microbial gas production of around 3% between 1850 and 2020 (Supplementary Fig. 14), resulting in additional emissions of 0.8 ± 0.4 Tg $N_2O$-N $a^{-1}$, and a consequent decrease in leaching losses. The parameterised 'warming' impact will be a combination of feedbacks driven by both changing precipitation and increased temperature, which cannot be separated in the current IsoTONE framework. The combined increase of 1.3 Tg $N_2O$-N $a^{-1}$ from spatial input changes and climate warming represents 18% of soil and 15% of total anthropogenic $N_2O$ emissions in 2020. As climate warming continues, and fertiliser use increases in many tropical and subtropical regions, the mean EF of $N_2O$ and thus the growth rate of tropospheric $N_2O$ will accelerate, unless significant efforts are focussed on $N_2O$ mitigation and increased fertiliser N use efficiency in emerging economies[2]. Moreover, the impact of potential unknown climatic feedbacks, in particular non-linear responses associated with

extreme events such as drought and flooding, should be a key focus for both measurement and model studies.

## Applications and outlook

$N_2O$ emissions over the past decades have increased strongly, following the trajectory predicted by the highest emission scenario (RCP8.5)[2], which suggests an increase in the global emission factor for $N_2O$. We developed and applied a coupled soil-atmosphere nitrogen isotope model framework—'IsoTONE'—using soil $\delta^{15}N$ as an emission proxy to understand the processes underlying spatiotemporal dynamics of global $N_2O$ emissions. This model set up developed in this study used isotopic composition to trace different N loss and $N_2O$ production processes, with a Markov Chain Monte Carlo approach implemented to constrain the model using tropospheric time series of $N_2O$ isotopic composition. Results from IsoTONE agree well with the CAMS inversion model[49], providing confidence in both methodologies. Compared to CAMS, IsoTONE is able to harness isotopic information to understand spatial variability in emission factors and production and loss processes; moreover, the simplified approach means IsoTONE has low computational requirements and can be used to explore questions requiring many simulations.

The model results show that fixation N inputs drive the majority of natural $N_2O$ emissions, but deposition N inputs account for the majority of anthropogenic emissions. Fertilisation N inputs are responsible for multidecadal variability in emissions, whereas subdecadal variability in $N_2O$ emissions is driven by biological N fixation. We show that the effective (N-input weighted) EF in 2020 is 4.3 ± 0.3%, much higher than the IPCC default value of 1.4%. $N_2O$ EFs are highly spatially heterogeneous—highest in warm, wet ecosystems—and thus strongly underestimated by default values based on area-weighted means. $N_2O$ EFs have increased over the past century, driven by climate warming as well as spatial redistribution of fertiliser N inputs. These two phenomena have led to additional emissions of 0.8 ± 0.4 and 0.5 ± 0.3 Tg $N_2O$-N $a^{-1}$ respectively between 1940 and 2020. Feedbacks between climate warming, spatial changes in agriculture, and $N_2O$ emissions should be considered in the development of emission projections and mitigation policies. Monitoring of annual $N_2O$ emissions as well as the soil $\delta^{15}N$ emission proxy in both understudied regions and regions with particularly high emission factors will be key to reduce uncertainty, focus mitigation efforts, and combat rising $N_2O$ emissions.

## Methods

### Soil, climate and N input datasets

Mean annual surface temperature (MAT) and precipitation (MAP) at 10 minute spatial resolution were taken from the Climate Research Unit high resolution global climatology dataset[77]. Global soil organic carbon estimates and topsoil (0–30 cm) bulk density were taken from the Harmonized World Soil Database with a spatial resolution of 30 arc seconds[78,79]. Aridity index at a spatial resolution of 30 arc seconds was taken from the Global Aridity Index and Potential Evapotranspiration Climate Database (v2)[80]. Global soil pH at >60,000 sites worldwide was taken from the database compiled by Slessarev et al.[81]. The Worldwide Organic Soil Carbon and Nitrogen Dataset[82] was used to estimate soil organic nitrogen content with data from >4000 sites rasterised using linear interpolation. Total fertiliser N inputs were taken from the Land Use Harmonization Database (LUH) with the historical dataset used for 1800–2015 (LUH2 v2h) and the future forcing dataset for 2015–2020 (LUH2 v2f)[9]. The Community Atmosphere Biosphere Land Exchange (CABLE) Australian community land surface model ([17,83], https://www.cawcr.gov.au/research/cable/) was used to estimate global mean water-filled pore space (WFPS), fractional $NH_3$ losses (no temporal variability) and deposition and fixation N inputs (annual)[17,83,84]. All datasets were converted to the model grid with 0.5° × 0.5° resolution for −180° to +180° longitude and −60° to +85° latitude (720 × 290

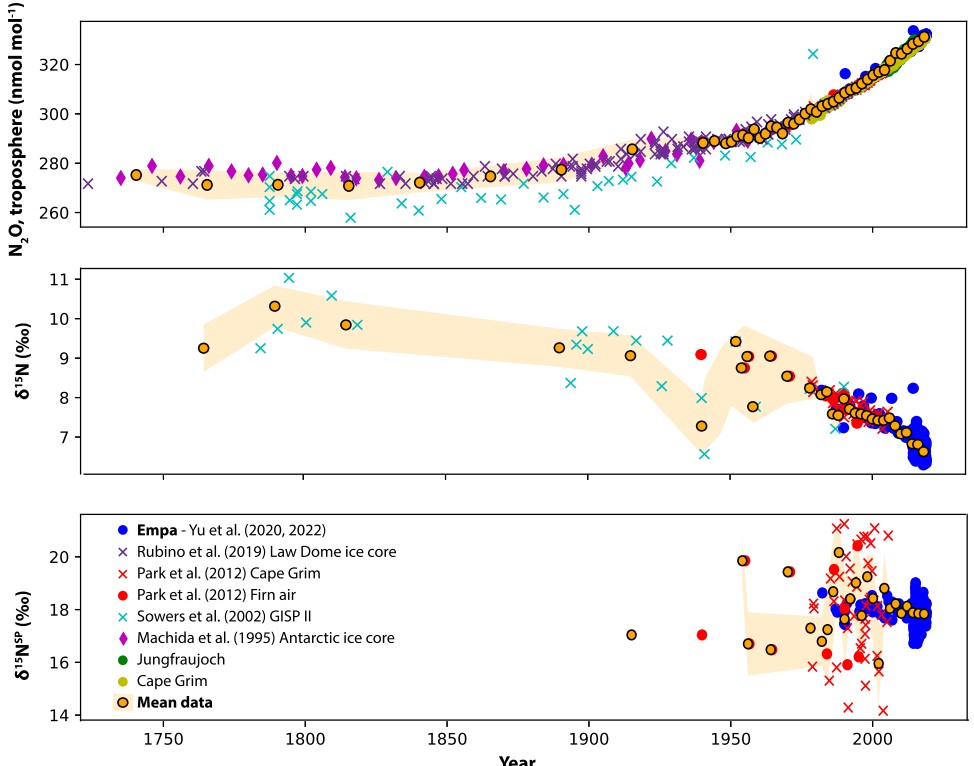

**Fig. 5 | Tropospheric background N₂O mixing ratio and isotopic composition ($\delta^{15}N$ and $\delta^{15}N^{SP}$) measured in different datasets.** Scale offsets between different datasets have been corrected using overlapping periods to match the Empa dataset as described in the text. Mean data at 25-year intervals (1740–1940) and 2-year intervals (1940–2018) are shown as orange points with black outline; the standard deviation of the data averaged for each point is shown as the orange shaded region.

gridcells) using the Python function `scipy.interpolate.griddata` with linear interpolation.

## Global dataset for $\delta^{15}N_{soil}$

Data for $\delta^{15}N_{soil}$ for >6000 soil samples from natural (non-agricultural) sites was compiled by Craine et al.[41]. This database also includes ancillary data such as MAT, MAP, carbon and nitrogen concentration, C:N ratio, soil texture, density and pH, and site coordinates and elevation, although not all parameters are available for all samples. Geographical coverage was improved by adding a further 748 samples[85-88], including unpublished data for 112 samples from Australia (BASE database;[89]) and 392 from Africa (experimental sites and set up described in refs. [90-96])—regions which were underrepresented in the Craine database. The majority of data was from near-surface soil (0–50 cm depth). Within this depth range, no significant effect of depth on $\delta^{15}N_{soil}$ was seen in the data. The sample-specific reported ancillary data was compared to the global gridded datasets (see Methods: Soil, climate and N input datasets) at the same geographical locations to gapfill the ancillary data (Supplementary Note 1). Relationships between ancillary data and $\delta^{15}N_{soil}$ measurements were then used to predict $\delta^{15}N_{soil}$ using an artificial neural network (ANN) with the Python package Keras ([97]; details in Supplementary Note 1). Using the ANN, global $\delta^{15}N_{soil}$ on an 0.5° × 0.5° grid was estimated in order to drive the soil isotope module (see Methods: Soil nitrogen module). A bootstrapping approach was used to estimate uncertainty in gridded $\delta^{15}N_{soil}$.

## Atmospheric data

A tropospheric background time series of N₂O mixing ratio and isotopic composition since the preindustrial era formed the core set of observational data used to constrain simulation results in this study (Fig. 5). The background mixing ratio, for the purposes of this study, is defined as the concentration of a given species when the impact of local or recent sources and sinks is absent—also known as the baseline concentration[98]. The primary dataset comprised tropospheric background measurements of N₂O mixing ratio, $\delta^{15}N$ and N-isotope $\delta^{15}N^{SP}$ from the Jungfraujoch High Alpine research station and the Cape Grim Air Archive (CGAA), as well as firn air sampled during the 2018 East GReenland Ice coring Project (EGRIP) campaign[45,99] covering the period from 1982 to the present day, all measured at Empa, Switzerland. Additionally, mixing ratio, $\delta^{15}N$ and $\delta^{15}N^{SP}$ data from CGAA (1979-2005) and firn air (1939-1995) from ref. [6] were used. These were corrected to the Empa dataset scale using the mean offset in the overlap period, with offsets of −1.1‰ and −1.0 ‰found for firn and CGAA data for $\delta^{15}N$, and +1.5‰ and +1.3‰ for firn and CGAA data for $\delta^{15}N^{SP}$. To extend the isotopic timeseries further back, ice core and interstitial snowpack air measurements from the Greenland Ice Sheet Project II (GISP II)[46] were also used, covering the period from 1785–1990. This dataset showed no offset for $\delta^{15}N$ compared to the Empa dataset, and did not include $\delta^{15}N^{SP}$ measurements. Other available datasets (e.g.[42,62]) were not integrated, as they would not have improved the temporal range of the combined dataset, and would have introduced additional calibration scale uncertainty.

N₂O mixing ratio in the preindustrial era was constrained using two Antarctic ice core datasets—from ref. [100] covering the period 1735–1964, and from ref. [101] covering the period from 154–1986. Additionally, measurements of N₂O mixing ratio available at the World Data Centre for Greenhouse Gases (WDCGG, https://gaw.kishour.go.jp) supported by the Global Atmosphere Watch (GAW) Program at Jungfraujoch (2005-present) and the Cape Grim Baseline Atmospheric Pollution Station (1980–present; ALE/GAGE/AGAGE program) were used. Cape Grim data comprise measurements from three instruments covering different time periods, so overlapping periods were used to correct scale offsets, with the most recent dataset (1993-2018) used as

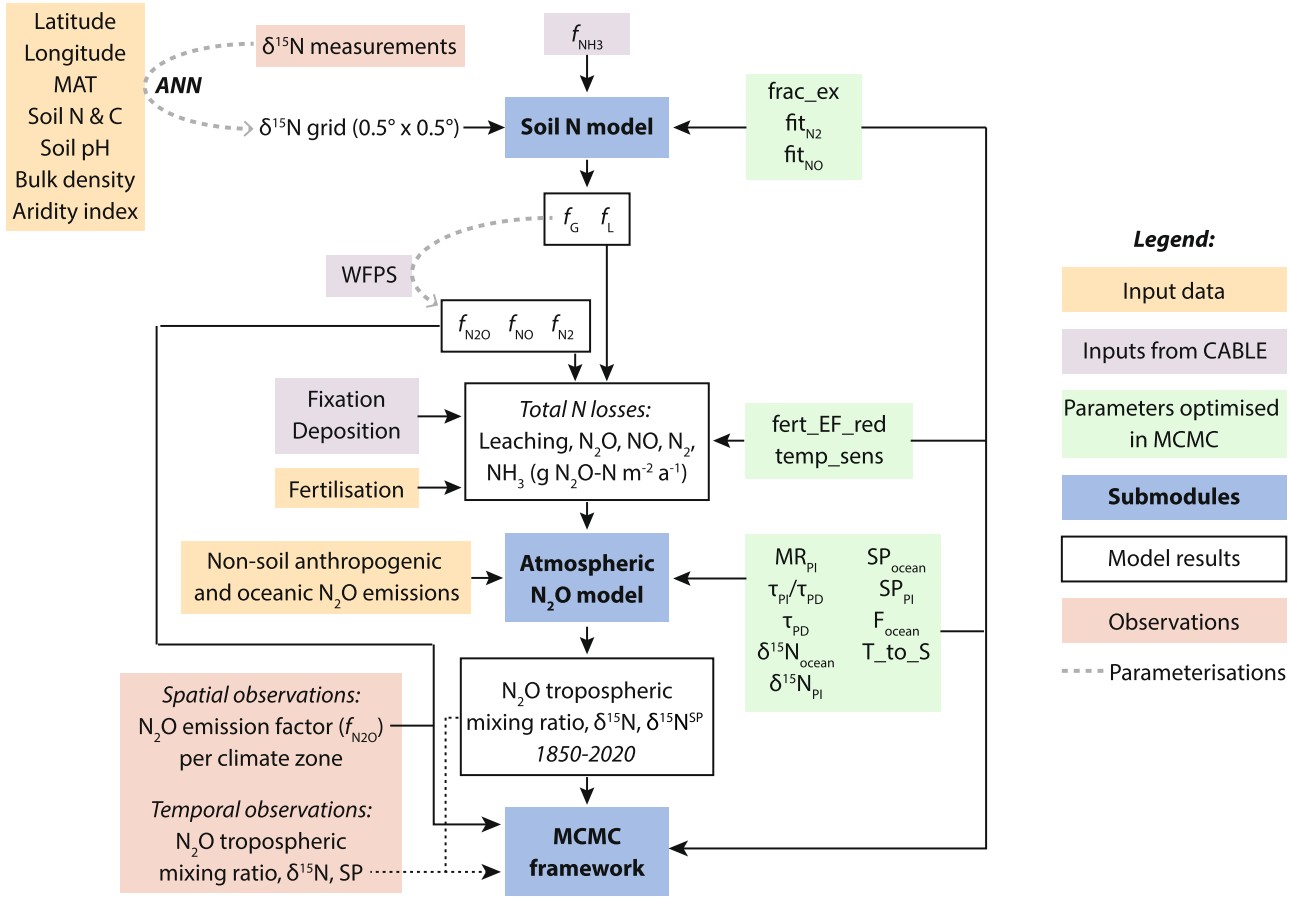

**Fig. 6 | A schematic view of the IsoTONE model and inversion structure, showing different input and output data types.** Optimized variables (green) are defined in Supplementary Table 1. Sources of input and observation data are given in the Methods . MAT Mean Annual Temperature, WFPS Water-Filled Pore Space, CABLE Community Atmosphere Biosphere Land Exchange land surface model, MCMC Markov Chain Monte Carlo; $f_G, f_L, f_{NO}, f_{N2O},$ and $f_{N2}$ refer to the proportion of N lost through leaching and N gas production, discussed in the Methods.

an anchor. The dataset from 1981–1994 was +0.03 nmol mol$^{-1}$ offset from the most recent dataset, and the dataset from 1978–1985 was −0.07 nmol mol$^{-1}$ offset from the 1981–1994 dataset.

To allow comparison to simulated results and minimize the impact of any strong episodic local sources, average values for mixing ratio and isotopic composition were binned into 25-year intervals (1740–1940) and 2-year intervals (1940–2018) (Fig. 5). Differences between northern and southern hemisphere sites were not significant given the measurement uncertainty, thus the troposphere was considered as a single well-mixed box to facilitate comparison with Iso-TONE simulations. The average standard deviation within each time window for 1740–1940 data was 5.4 nmol mol$^{-1}$, 0.5‰ and 1.2‰ for mixing ratio, $\delta^{15}$N and $\delta^{15}$N$^{SP}$, compared to 0.8 nmol mol$^{-1}$, 0.2‰ and 0.6‰ for mixing ratio, $\delta^{15}$N and $\delta^{15}$N$^{SP}$ for 2000–2020, reflecting both the strong improvement in measurement techniques for recent in situ data and the particular challenges of firn and ice core measurements. The trend in mixing ratio for averaged data over the last 40 years was +0.8 nmol mol$^{-1}$ a$^{-1}$, with trends of −0.03‰ a$^{-1}$ and −0.006‰ a$^{-1}$ observed for $\delta^{15}$N and $\delta^{15}$N$^{SP}$ respectively.

## N$_2$O fluxes and emission factors
Data for N$_2$O fluxes and emission factors (EF) for >300 sites has been compiled in the 'Global N$_2$O Database'[102]. This database also includes ancillary data such as MAT, MAP, carbon and nitrogen concentration, C:N ratio, soil texture, density and pH, and site coordinates and elevation, although not all parameters are included for all samples. The sample-specific reported ancillary data was compared to the global gridded datasets at the same geographical locations (see Supplementary Note 1). The strongest relationships between ancillary data and emission factors were determined, and used to bin emission factor data into 16 different geographical regions to simplify comparison with modelled emission factors and minimise the impact of localised emission hotspots (see details in Supplementary Note 1: N$_2$O emission factors).

## Coupled soil-atmosphere model
The model used in this study is an extension of the simple soil isotope model presented by Bai et al.[39], coupled to an extended version of the atmospheric box model described in ref. 45 to simulate gridded N$_2$O fluxes as well as tropospheric mixing ratio and isotopic composition from 1800 to 2020. The full coupled model was written in `Python` and will be hereafter referred to as the IsoTONE model (ISOtopic Tracing Of Nitrogen in the Environment). A schematic of the model is shown in Fig. 6 and a detailed description is given in the following subsections.

## Soil nitrogen module
The soil N module of IsoTONE estimates fractional losses of N to different pathways based on $\delta^{15}$N$_{soil}$ (see Methods: Global dataset for $\delta^{15}$N$_{soil}$). The module is based on the equations used by Houlton, Bai[38] and Bai et al.[39], which state that the $\delta^{15}$N$_{soil}$ reflects the balance between N inputs ($i$), with a relatively constant rate and isotopic composition, and variable N losses. Loss pathways are divided into leaching ($L$), with low isotopic fractionation; NH$_3$ volatilization, with high isotopic fractionation but a low contribution to total losses ($NH_3$); and abiotic and

microbial gas production ($G$), with high and variable isotopic fractionation. At steady state, inputs must equal outputs for both total amounts and isotopic composition:

$$f_L + f_{NH3} + f_G = 1 \qquad (1)$$

$$\delta^{15}N_{soil} = \delta^{15}N_i - \varepsilon_G \times f_G - \varepsilon_L \times f_L - \varepsilon_{NH3} \times f_{NH3} \qquad (2)$$

where $\varepsilon$ is the fractionation factor ($k_{15_N}/k_{14_N}$ expressed in permil, where $k$ is the reaction rate) for the respective loss pathway and $\delta^{15}N_i$ was set as $-1.5‰$[39]. Assuming steady state, the fraction of N loss accounted for by leaching ($f_L$), NH$_3$ volatilization ($f_{NH3}$) and gas production ($f_G$) can be calculated from $\delta^{15}N_{soil}$. The steady state assumption is only valid for natural ecosystems, thus only $\delta^{15}N_{soil}$ measurements from natural sites were used to initialise the model. Unlike[39], $^{14}$N and $^{15}$N were traced separately in Eq. (2), whereby the rate of $^{15}$N loss via a pathway is the rate of $^{14}$N multiplied by the fractionation factor, for example, for leaching:

$$k_L^{15} = k_L^{14} \times \left( \frac{\varepsilon_L}{1000} + 1 \right) \qquad (3)$$

The rate of $^{15}$N addition was calculated analogously. Tracing isotopes separately facilitated calculation of the stepwise impact of N losses on the remaining N reservoir isotopic composition ('Rayleigh fractionation'[103]), and necessitated an iterative solution to Eq. (2). Rayleigh fractionation describes how the isotopic composition of reactant and product pools change as a reaction proceeds depending on the fractionation factor and the fraction of reactant consumed, assuming both reactant and product pools are well mixed[103].

We assumed that N loss processes are primarily linked with the inorganic N pools (i.e. NO$_3^-$ and NH$_4^+$), and estimated fractionation factors related to the two species. Fractionation during N mineralization was assumed to be minor[104]. Soil organic N was not considered, given that the rates of turnover processes and their associated isotopic fractionation effects are very scarcely studied at the global scale. Fractionation during leaching ($\varepsilon_L$) was estimated to be 1‰[39]. Fractionation during NH$_3$ volatilization ($\varepsilon_{NH3}$) was estimated to be $-17.9‰$ based on the equation in ref. 105. Fractionation during N gas production ($\varepsilon_G$) was estimated based on the relative contributions of the two dominant microbial processes: (i) Fractionation for N$_2$O production during nitrification of NH$_4^+$ ($\varepsilon_{nit}$) was estimated as $-56.6 \pm 7.3‰$ with $\delta^{15}N^{SP}$ of $29.9 \pm 2.9‰$[106]. (ii) Fractionation for N$_2$O production during denitrification of NO$_3^-$ ($\varepsilon_{denit}$) was estimated by adding the fractionation factor for reduction of NO$_3^-$ to NO$_2^-$ ($-31.3 \pm 6.1‰$) and for $rmNO_2^-$ to N$_2$O ($-14.9 \pm 6.7‰$)[106], thus $\varepsilon_{denit} = -46.2‰$. $\delta^{15}N^{SP}$ for N$_2$O produced from denitrification was estimated as $-1.6 \pm 3.0‰$ (iii) Fractionation for N$_2$O reduction during denitrification ($\varepsilon_{red}$) was estimated as $-6.6 \pm 2.7‰$ with $\delta^{15}N^{SP}$ of $-5 \pm 3‰$[106].

In contrast to bacterial and abiotic denitrification, fungal denitrification results in N$_2$O with a high and variable $\delta^{15}N^{SP}$. The partitioning of N$_2$O production into fungal denitrification as compared to bacterial nitrification and denitrification (shown in Supplementary Fig. 6) is currently not known and therefore cannot be parameterised within this model. Fungal denitrification is a minor source of N$_2$O, expected to contribute less than 10–15% globally[107]. Similarly, co- and chemodenitrification are poorly constrained pathways, both in terms of drivers and isotopic composition[44,108,109]. Fungal denitrification and co- and chemodenitrification will be included in a future version of this model as data to drive parameterisations becomes available.

These fractionation factors refer to the immediate production or consumption of N substrates and N$_2$O, and thus do not account for the complexity of processes leading to the net isotopic composition of emitted N$_2$O in real environments. Several studies have recognised

that soil gas is not a homogeneous, well-mixed environment—instead, gas production and consumption occurs in pores that are only partially connected relative to the rate of microbial processes[110,111]. This means that Rayleigh fractionation processes occur within all pores and microenvironments, and thus the effective net fractionation may be significantly lower than calculated from measured fractionation factors, often referred to as 'underexpression'[110]. We therefore introduce a parameter called the fractionation expression factor (frac_ex) to the model such that net or effective fractionation = $\varepsilon \times$ frac_ex for all modelled processes. Thereby, frac_ex = 1 would reflect a well-mixed soil gas reservoir where effective fractionation is equal to expectations from laboratory measurements, and frac_ex = 0 would reflect a completely closed soil gas environment where all reactions go to completion within separate pores and thus no effective fractionation is observed. The initial value of frac_ex was set to 0.7 and it was optimized in the MCMC framework (see Methods: Optimization of model N cycle parameters) assuming a uniform error distribution between 0.3 and 1.0. In reality, different values for frac_ex may be expected for different processes and environments, in particular depending on soil texture and pore structure, however this cannot be determined within the available model-data framework.

The soil model was run on an $0.5° \times 0.5°$ grid from $-60°$ to $80°$ latitude and $-180°$ to $180°$ longitude. The $\delta^{15}$N grid (see Methods: Global dataset for $\delta^{15}N_{soil}$; Supplementary Fig. 2a) was first initialised in each model run by adding 5% of the uncertainty in $\delta^{15}$N (Supplementary Fig. 2b) multiplied by a normally distributed random number ($\mu = 0$, $\sigma = 1$). For each grid cell, the soil model was initialised with an available soil nitrogen pool of size 1 (unitless) with $\delta^{15}$N of 0‰, and a soil N$_2$O pool of of size 1 (unitless) with $\delta^{15}$N and $\delta^{15}N^{SP}$ of 0‰. The $^{14}$N input rate ($k_i$) was set at 1 (unitless) and the $^{15}$N input rate ($k_{i,15}$) calculated using $\delta^{15}N_i$ of 0.5‰ (based on[39,112–114]). Changing the unitless pool sizes affects only the number of iterations until steady state is reached in the model, and not the final result of the model. Reducing $\delta^{15}N_i$ by 1‰ increased the calculated mean global $f_G$ by -1%, while increasing $\delta^{15}N_i$ by 1‰ increased the calculated mean global $f_G$ by -5%, with very little impact of spatial distribution of $f_G$.

To solve Eq. (2) for each grid cell, the soil module was run over four iterations ($n = 4$). For the first iteration, $f_G$ was set at 0.1 for all gridcells. $f_{NH3}$ was estimated using the CABLE model[17,83]. $f_{NH3}$ is nearly always <0.05, therefore the model results are not highly sensitive to the parameterisation of $f_{NH3}$. $f_L$ was then determined as $1 - f_G - f_{NH3}$. Partitioning of total gas losses ($f_G$) into N$_2$O, N$_2$ and NO ($f_{N_2O}, f_{NO}, f_{N_2}$) was estimated with a sigmoid fit of WFPS to available experimental measurements (Supplementary Note 2, Supplementary Fig. 5), with mean global WFPS from CABLE used to constrain partitioning for each grid cell in the model. Similarly, denitrification ($f_{denit}$) and nitrification ($f_{nit}$) contributions to N$_2$O production were estimated with a sigmoid fit to available experimental data (Supplementary Note 2, Supplementary Fig. 6). Neither WFPS nor other potential proxies such as soil oxygenation can adequately describe the microenvironment in which microbes conduct N cycling[44,115]. WFPS represents the amount of pore space filled with water and with air and can be compared between soils with different textures, thus it can be used as a proxy for the ability of gases and substrates to move through aqueous and gaseous environments, which is key in determining both substrate supply and soil oxygenation, and thus N cycling and gas production. Mean WFPS will not describe all variability in gas partitioning, given the highly variable and non-linear nature of N gas emissions, however it provides the best available estimate based on the current status of experimental and modelling research. An overall $\varepsilon_G$ was estimated based on $f_{denit}$ and $f_{nit}$ and the fractionation factors for the individual processes multiplied by frac_ex, with a mean value of $-30‰$. This is lower than in ref. 39 where a range of $-16$ to $-20‰$ was assumed for $\varepsilon_G$ without explicit consideration of microbial pathways—measurements made since the publication of ref. 39 have shown that fractionation is much larger than previously

estimated[106]. To achieve steady state, 10 cycles of N addition and removal were conducted within each iteration. First, N inputs for $^{14}$N and $^{15}$N were added to the initial soil N pool. N was then removed via leaching, volatilization, and gas production for $^{14}$N and $^{15}$N. The steady state soil pool $\delta^{15}$N value at the end of each iteration ($\delta^{15}$N$_{ss}$) was compared to the $\delta^{15}$N value for the gridcell ($\delta^{15}$N$_{soil}$) to find an improved $f_G$ for the next iteration according to:

$$f_{G,n+1} = f_{G,n} - \frac{\delta^{15}N_{soil} - \delta^{15}N_{ss}}{\varepsilon_G} \tag{4}$$

Following four iterations, $\delta^{15}$N$_{soil}$ and modelled $\delta^{15}$N$_{ss}$ agreed within 0.01‰ and final values of $f_G$ and $f_L$ were accepted.

In addition to the calculation of $f_G$ and $f_L$, the soil module estimated N$_2$O production rate and isotopic composition for each grid cell. The production rates for $^{14}$N-N$_2$O and $^{15}$N-N$_2$O were calculated from the fractionation factors for each pathway multiplied by frac_ex; both $^{15}$N$^\alpha$ and $^{15}$N$^\beta$ were traced separately to consider N$_2$O $\delta^{15}$N$^{SP}$. For example, the $^{14}$N reaction rate for nitrification N$_2$O production was estimated as $f_G \times f_{N_2O} \times f_{nit}$ and the corresponding reaction rate for $^{15}$N as:

$$k_{15N,nit} = f_G \times f_{N_2O} \times f_{nit} \times \left( \frac{\varepsilon_{nit} \times frac\_ex}{1000} + 1 \right) \tag{5}$$

Analogously, rates for each isotopic variant were also calculated for denitrification production and reduction. The production of N$_2$ was assumed to represent the consumption of N$_2$O via complete denitrification, as no other significant N$_2$ sources are known, thus the extent of N$_2$O reduction could be estimated and its isotopic effect calculated. N$_2$O production and consumption were estimated in each step of the steady state calculation to give a final N$_2$O flux and isotopic composition ($\delta^{15}$N and $\delta^{15}$N$^{SP}$) for each gridcell to pass on to the atmospheric module described in the following subsection.

**Atmospheric N$_2$O module**

The atmospheric module takes the fractional losses, estimated by the soil module described in the previous subsection, and uses a two-box model representing a well-mixed troposphere and stratosphere to reconstruct N$_2$O background mixing ratio and isotopic composition. Absolute N emissions into the tropospheric box were calculated using the fractional estimates of losses for each grid cell from the soil module, combined with annual N inputs to each grid cell for fixation, deposition and fertilisation for the period 1800–2020. Fertiliser N inputs were taken from the LUH database and covered the whole time period (see Methods: at the same geographical locations to gapfill the ancillary data). Deposition N inputs for 1860–2050 were from ref. 116. Fixation (for 1901–2100) was estimated using the CABLE model with scenario A1 as described by Peng et al. [17], where fixation is calculated using resource optimization with temperature dependence[84]. The 1901 fixation values were used to estimate fixation in earlier years and the 1860 values for deposition in earlier years, thus assuming negligible anthropogenic influence before 1850. As none of the N input datasets included significant anthropogenic inputs before 1850, 1850 is taken as the preindustrial 'baseline' throughout this study.

Deposition and fixation inputs were assumed to generally fulfil the steady state criteria required by the soil module, however harvest N exports mean that this assumption is not valid for fertiliser N inputs. Harvested N is often returned to the soil or atmosphere via manure or deposition at a different location[117]. Previous studies have shown that a large proportion of fertiliser N is incorporated into crops and therefore removed during harvest, as well as potentially stored in soils ([10,52,53,117] and references therein)—thus not available for 'normal' N loss partitioning. Therefore, we added a factor fert_EF_red to the model to account for N that is harvested or stored. Fertiliser N inputs are

multiplied by fert_EF_red to account for the removal of harvested N or stored in soils before losses by other pathways are calculated; fert_EF_red of 0 would mean that all N is removed in harvest, and fert_EF_red of 1 would mean no N is removed. The isotopic impact of harvest N exports cannot be accounted for within the scope of this study, as the required input data (in particular the proportion of N removed in harvest per grid cell annually) is not currently available. Harvest N will be explicitly incorporated into a future version of the IsoTONE framework. EF changes with fertilisation rate were not considered: A recent meta-analysis showed that the dependence of EF on fertilisation rate is highly variable, and at the global scale EF is not strongly dependent on fertilisation rate[118], although some studies have shown a strong non-linear relationship between EF and N inputs at the site and regional scale[119]. Tian et al.[2] report that recent increases in EF for direct soil emissions are likely due to climate change feedbacks and interactions, rather than a direct increase in EF due to increasing fertiliser application. As more evidence regarding the non-linearity of fertiliser EFs becomes available, this will be incorporated into the IsoTONE model.

Previous studies have shown that microbial N gas production is temperature sensitive[23,48,120] and likely increasing in a warming climate by between 0.5 and 1.0 Tg N$_2$O-N a$^{-1}$ °C$^{-1}$, with a best estimate of 0.6 ± 0.2 Tg N$_2$O-N a$^{-1}$ °C$^{-1}$[18,33,121]. We assume that this increase in emissions relates to a general sensitivity of microbial activity to temperature, and is therefore likely to affect production of N$_2$ and NO as well as N$_2$O. We therefore incorporated a temperature sensitivity increase of 10% of microbial N emissions from the year 1800 per degree of warming (temp_sens = 1.1 ± 0.04) for N$_2$O, N$_2$ and NO, which corresponds to 0.6 Tg N$_2$O-N a$^{-1}$ °C$^{-1}$ according to the best estimate of 6.3 ± 1.1 Tg-N a$^{-1}$ for N$_2$O emissions in the preindustrial era[8]. Temperature anomalies from the CRUTEM-4.6.0 dataset were used to drive the temperature sensitivity calculation[122]. N losses through leaching were consequently reduced to maintain N mass balance.

Yearly soil emissions of N$_2$O, NO and N$_2$ were calculated for each input type for each grid cell incorporating both fert_EF_red and temperature sensitivity of emissions. In addition, EDGAR (Emission Database for Global Atmospheric Research,[123,124]) gridded emissions for the major non-soil anthropogenic N$_2$O emission categories (1A1 = power industry, 1A3b = road transport, 2B = chemical processes, and 6 = wastewater treatment) were added to estimate total terrestrial N$_2$O emissions. Isotopic composition was estimated as $\delta^{15}$N = 3.9 ± 2.9 and $\delta^{15}$N$^{SP}$ = 17.6 ± 0.5 for power industry, $\delta^{15}$N = −7.2 ± 1.2 and $\delta^{15}$N$^{SP}$ = 10.0 ± 4.3 for road transport, $\delta^{15}$N = −8.3 ± 10.6 and $\delta^{15}$N$^{SP}$ = 3.3 ± 5.5 for chemical processes, and $\delta^{15}$N = −11.6 ± 12.7 and $\delta^{15}$N$^{SP}$ = 10.0 ± 5.7 for wastewater treatment[43]. Isotopic composition of natural soil N$_2$O emissions was given by the soil module for each grid cell, and weighted by the emissions per grid cell to calculate overall isotopic composition of soil N$_2$O emissions. The $\delta^{15}$N isotopic composition of fertiliser N inputs is estimated to be 3‰[125,126] compared to −1.5‰ for natural N inputs[39], thus increasing the $\delta^{15}$N of emitted N$_2$O. Total global emissions were found by adding the oceanic emissions (F$_{ocean}$) to the global terrestrial emissions using the flux and isotopic compositions listed in Supplementary Table 1.

Total global N$_2$O was emitted into a two-box atmospheric model comprising a well-mixed troposphere and a well-mixed stratosphere, based on the model described in ref. 45. Emissions were assumed to be immediately well-mixed through the atmosphere, as the time required for tropospheric mixing is estimated to be around one year[127], which is the time resolution of the model. Previous versions of this model explicitly calculate the required preindustrial terrestrial N$_2$O flux to achieve steady state in the preindustrial era accounting for best estimates of tropospheric N$_2$O mixing ratio, N$_2$O lifetime, and stratosphere-troposphere exchange. However, in the atmosphere module of IsoTONE, preindustrial terrestrial emissions (F$_{terr}$) are prescribed by the soil module. An iterative calculation (maximum of 20

iterations) is therefore used to optimize the preindustrial oceanic $N_2O$ flux ($F_{ocean}$) and the troposphere-stratosphere exchange term ($T\_to\_S$) by shifting these two terms stepwise according to their prior uncertainty (Supplementary Table 1) using two equations until steady state is achieved. First, tropospheric $N_2O$ inputs are balanced against losses to the stratosphere:

$$F_{terr} + F_{ocean} = T\_to\_S \times (MR_{PI} - MR_{PI,strat}) \qquad (6)$$

where $MR_{PI}$ and $MR_{PI,strat}$ are the $N_2O$ mixing ratios in the preindustrial troposphere and stratosphere respectively. The preindustrial stratospheric $N_2O$ mixing ratio at steady state is then calculated based on stratosphere-troposphere exchange and stratospheric $N_2O$ destruction[42,63]:

$$MR_{PI,strat} = -\frac{MR_{PI} \times moles_{trop} \times MW_{N2O-N} - T\_to\_S \times MR_{PI} \times MW_{N2O-N} \times \tau_{PI}}{T\_to\_S \times MW_{N2O-N} \times \tau_{PI} + moles_{strat} \times MW_{N2O-N}}$$

$$(7)$$

where $MW_{N2O-N}$ is the molecular weight of N in $N_2O$ (28 g mol$^{-1}$) and $moles_{trop}$ and $moles_{strat}$ are the moles of air in the troposphere and the stratosphere (1.5 and $0.27 \times 10^{20}$ moles respectively). This iteration meant that optimized values for both $F_{ocean}$ and $T\_to\_S$ were found for each solution of the model, although they were not explicitly targeted in the inversion described in the following subsection. $F_{ocean}$ was not varied with time, as recent results suggest that the oceanic flux is relatively stable compared to the terrestrial flux[2]. Smaller, dynamic aquatic and semiaquatic ecosystems, such as estuaries and coastal wetlands, show highly dynamic and variably N cycling and $N_2O$ emissions[128–130], but are too small to be resolved at the global scale of this study.

Once steady state was achieved for preindustrial $N_2O$ fluxes and mixing ratio, model calculations proceeded as described by Yu et al.[45] and will therefore only be briefly presented here. First, net stratospheric fractionation and the isotopic composition of the preindustrial stratosphere were found for $\delta^{15}N$ and $\delta^{15}N^{SP}$ assuming steady state. The model was then run forwards with annual time steps using annual terrestrial emissions and isotopic composition provided by the soil module of IsoTONE. At each time step, $N_2O$ inputs and destruction were considered to estimate the rate of change in $N_2O$ mixing ratio and isotopic composition, and thus calculate a time series of mixing ratio, $\delta^{15}N$ and $\delta^{15}N^{SP}$ for a well-mixed troposphere[45].

### Optimization of model N cycle parameters

Twelve key model parameters were optimised using a Markov Chain Monte Carlo (MCMC) approach (Supplementary Table 1), with several datasets used to constrain the results. The geoclimatic gradients (spatial variability) were constrained using $N_2O$ emission factors from the Global $N_2O$ Database, binned for 16 climatic zones (see Methods: $N_2O$ fluxes and emission factors). Temporal variability was constrained using a combined background tropospheric timeseries of $N_2O$ mixing ratio and isotopic composition ($\delta^{15}N$ and $\delta^{15}N^{SP}$) from several different sites, averaged for 25-year blocks from 1740 to 1940 and for 2 year blocks from 1940 to 2020 to give a total of 49 data points (see Methods: Atmospheric data and Fig. 5). Data was reduced to 16 climate zone EFs and 49 temporal data points to avoid strong overweighting of recent atmospheric results, which are much more frequent and less uncertain than older measurements (e.g. biweekly monitoring at Jungfraujoch station[45]) but highly covariable. Observation uncertainty was defined as the standard deviation of measurements within each spatial or temporal block. Model uncertainty was set at 0.5 nmol mol$^{-1}$ for $N_2O$ mixing ratio, 0.1‰ for $\delta^{15}N$ and $\delta^{15}N^{SP}$, and 0.5 for climate zone EF. The incorporation of isotopic composition in the atmospheric model gave an implicit sensitivity to the spatial distribution of emissions, as the isotopic composition of emitted $N_2O$ depends on the

dominant emission processes in a particular gridcell, thus allowing the model to distinguish between changes in different regions and input types.

The MCMC was run with three different stepsizes: 0.75, 0.5 and 0.25. Within each iteration $i$ of the MCMC, parameters following a Gaussian uncertainty distribution (see Supplementary Table 1), were varied according to:

$$P_{i+1,G} = P_{i,G} + 1\sigma \text{ uncertainty} \times \text{stepsize} \times r_{unif} \qquad (8)$$

where $P_{i,G}$ is the value of the Gaussian parameter in iteration $i$ and $r_{unif}$ is a uniformly distributed random number between -1 and 1. Uniform parameters were varied according to:

$$P_{i+1,U} = P_{i,U} + \frac{R_{max} - R_{min}}{4} \times \text{stepsize} \times r_{unif} \qquad (9)$$

where $P_{i,U}$ is the value of the uniformly-distributed parameter in iteration $i$, and $R_{max}$ and $R_{min}$ are the maximum and minimum of the parameter uncertainty range. Independent values of $r_{unif}$ were determined for every parameter. Observation uncertainty follows a Gaussian distribution, thus observations were also varied within each iteration using Eq. (8), however $r_{unif}$ was determined separately only for different groups of observations, e.g. $N_2O$ mixing ratio, $N_2O$ $\delta^{15}N^{SP}$.

Once parameters were varied within an iteration $i$, the Metropolis rule was applied to determine if parameters and observations could be accepted[131]. If both were accepted, the model was run for the parameter set $i$, and the model-observation probability was calculated for $i$. The Metropolis rule was then applied to determine if model-observation $i$ could be accepted compared to $i-1$. 5000 iterations were run at each step size sequentially until a total of 120,000 iterations had been run (40,000 at each step size), which was sufficient to achieve stable results with no significant difference between posterior parameters by step size, and no change in posterior parameters following more iterations. All tested and accepted parameter sets were saved; tested parameter sets were used to check coverage of the parameter uncertainty space (Supplementary Fig. 7), and accepted parameter sets were used to find posterior results and uncertainties for the parameters (Supplementary Table 1). The parameters $F_{ocean}$ and $T\_to\_S$ (see Methods: Atmospheric $N_2O$ module) were not explicitly varied in the MCMC but were calculated to achieve steady state in the atmospheric module in each accepted iteration, thus posterior estimates for these parameters were also obtained.

### Estimating uncertainty in posterior model results

The uncertainty in posterior parameters shown in Supplementary Table 1 was estimated as the standard deviation of all accepted results. To estimate uncertainty in the posterior model results, 100 iterations of the model were run using 100 randomly selected sets of accepted parameters. The standard deviation of results from all 100 iterations was used to estimate the uncertainty in the final model results. Standard error propagation was used to estimate uncertainty in all subsequently calculated values, eg. ratios and sums across time or space.

### Data availability

The gridded input datasets generated in this study have been deposited in the public, open access model code repository: https://github.com/elizaharris/IsoTONE. $\delta^{15}N_{soil}$ point data not present in the[41] dataset are also included in this repository, and also archived in the Pangaea data repository (doi currently being processed). Tropospheric background $N_2O$ isotopic data collected at Empa and used for model

optimization will be released open access in 2022 together with the associated manuscript[99].

## Code availability

The full IsoTONE model code including implementation of the MCMC optimization is included in the public, open access repository: https://github.com/elizaharris/IsoTONE. The repository doi for the associated release IsoTONE_v1 is 10.5281/zenodo.6772207.

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

## Acknowledgements

This work was supported by the Austrian Science Foundation (FWF) project P31132 'NitroTrace: Using isotopes to trace the effects of climate extremes on $N_2O$ emissions and the nitrogen cycle in managed grasslands' and the Swiss National Science Foundation (SNF) project 163075 'Assessment of the global $N_2O$ budget based on seasonal and long-term isotope measurements at Jungfraujoch and the Cape Grim Air Archive'. An OECD Cooperative Research Program for Sustainable Agricultural and Food Systems (OECD-CRP) grant for the project 'Identifying drivers of $N_2O$ emissions in a changing climate' supported Eliza Harris' research stay at the University of Melbourne to carry out this work. Longfei Yu was supported by the EMPAPOSTDOCS-II program, which receives funding from the European Union's Horizon 2020 research and innovation program under the Marie Sklodowska-Curie grant agreement number 754364.

The $N_2O$ observations at Jungfraujoch were supported by the International Foundation High Altitude Research Stations Jungfraujoch and Gornergrat (HFSJG), the Swiss Federal Office for the Environment, and ICOS-CH (Integrated Carbon Observation System Research Infrastructure), and we would like to thank Christoph Zellweger (Empa) for assistance with data collection from these sites. We would also like to thank Eric Slessarev (Lawrence Livermore National Laboratory) for provision of soil pH data, and Kate Summer and Matheus Carvahlo de Carvahlo (Southern Cross University) for assistance with sample preparation and data collation for Australian soils in the BASE database. The inverse modelling results were provided by R. L. Thompson who was funded through the Copernicus Atmosphere Monitoring Service (https://atmosphere.copernicus.eu/), implemented by ECMWF on behalf of the European Commission, and were generated using computing resources from LSCE.

## Author contributions

E.H. and P.R. conceptualised the study, with guidance from M. Bahn. and Y-P. W. based on the earlier work of E.B. E.H. wrote the model code, ran the model, and wrote the manuscript. P.R., S.H. and M.R. guided the data assimilation and machine learning. L.Y . and J.M. led the measurement of tropospheric background $N_2O$ isotopic composition; P.B.K. and Z.M.L. provided samples and concentration data for the Cape Grim station. M. Barthel, M. Bauters, P.B., M.F., J.S. and N.S.W. provided soil $\delta^{15}N$ data, and S.H., C.D. and M.S. provided other input data. All coauthors reviewed and contributed to the final version of the manuscript.

## Competing interests

The authors declare no competing interests.
