## [Peer Review File · Nature Communications]

Warming and redistribution of nitrogen inputs drive an increase in terrestrial nitrous oxide emission factorREVIEWER COMMENTS

Reviewer #2 (Remarks to the Author):

The manuscript presents an extensive body of work on modelling N₂O emissions using delta 15N isotopic approaches. The explanation of the methodology was extensive, and the findings well presented. The work improves our understanding of N₂O emissions globally and will have important consequences for nitrogen management. I did find the manuscript to be long, primarily as a consequence of the extensive supplementary information, and 13 figures and 1 table in the supplementary material. On a first read of the manuscript, it was quite difficult to keep track of all the figures and data, and the presentation of the supplementary figures did not seem to follow a logical (chronological) order in the manuscript. For example, Figure A.12 is the 1st figure referenced in the manuscript (line 146). I wonder if it is possible to present the data in a manner that does not require continuous reference to the supplementary figures for the reader to more readily comprehend the information.

I was also a little unclear on the focus of the manuscript. The title and much of the discussion is about the predicted emissions and reasons for any changes from 1850 to 2020. So is the focus about the ability of the model to predict N₂O emissions or about N₂O emissions predicted by the model? This might affect the arrangement of the presented figures.

Specific comments

Please clarify terminology EF. Is this emission factors or emission factor? Please check throughout the manuscript.

Line 146. Figure A.12 is referenced for terrestrial N₂O emissions and comment is made on emissions in 1860 and 2010. However Figure A.12 only shows the 2020 data. Also in line 157, reference to the year 2020 should be made.

Figure A.12 mentions anthropogenic emissions as being the increase above those in 1800 whereas elsewhere in the manuscript 1850 is referred to as the baseline (e.g. line 223, Figure A.11). Can the authors clarify what the baseline year is in the modelling of anthropogenic emissions?

Table 1. Please make clear the meaning of 'Growth rate is the 10-year mean growth rate centred on the year of interest.' Does this mean the 10 year average for 1850 is the average from 1845-1855 (as 1850 is the centre of that) or from 1840 to 1850?

Table 1 should be placed before Figure 1 according to the order of citing in the manuscript

Line 144-145. Reference is made to the model outputs agreeing with previous results. Are the previous results from the literature and if so, please add citations. If not from the literature, can these previous results be shown?

Table A1. The term PD is being used for present day (i.e. in description of 'Ratio of N₂O lifetime...' in unitless and years descriptions) and probability distribution (last column of table). Please modify one or other to make this clear.

Table A1. Terminology is not defined. For example, Table A1. SPOcean is defined as Mean SP for the ocean source but SP should be defined also. Pre-industrial (PI) should be defined in the 1st row of data (MRPI) not the 5th row

Line 884 refers to Section A.3 and is in Section A.3, suggest delete

How is the Isotone model better than the CAMS inversion model?

Figure A10. Chart e). It is not clear how the proportion of N₂O emissions (%) from nitrification and denitrification both reduce from 1800s. In 1800's this added makes 100% (40+60) but in 2020 the total is only 90%. What pathway is producing the remaining 10% of N₂O emissions? Caption states percentage for charts d) and e) but y-axis states proportion. Please use consistent terminology.

Fert_EF_red (Table A1) is a little confusing until reading lines 559-581. As this is a factor, I wonder if it should be written as EF reduction for fertiliser emissions factor (then matches the fractionation expression factor). I also think it would be helpful to point the reader to the section where this (and other terms in the Table) are defined.

Figure 2 2nd panel and Figure A.11 1st panel seem to be presenting the same data but the values are different on the y-axis although the caption seems to state the same assumptions. Can the authors make clear what the difference is between these 2 figures? Is it because the A.11 figure is based off estimates using 15N and Figure 2 uses non-labelled findings?

Figure 2 2nd panel and Table 1 data do not seem to match. Even though Table 1 presents the 10 year data, the values provided for fixation N increase from 0.2 to 1.8 whereas Figure 2 shows a value that looks stable at around 4. And the total anthropogenic soil emissions in Table 1 increase from 1 to 7 but in Figure 2 appear to increase from 5 to 10.

Figure 2. The scale starts at 1900 rather than prior to 1850 as shown in Figure A.11 and in Table 1.

Line 210 The authors state the anthropogenic N₂O flux in 2020 is 7.1 Tg per annum, but Figure 2 indicates a total of around 10 Tg per annum.

Figure 2 panel 3 has more lines of data (8) than provided in the legend (6) and the 3 year mean and 10 year mean colours are not visible in the chart.

Line 237 – does the Figure 2 panel 3 y-axis scale reach 0.3? It is difficult to see

Sometimes 'a-1 a-1' is used and sometimes 'a-2', should only 1 of these be used?

Figure 3 should be moved further down the manuscript to after line 274.

Line 286-287. I have a little trouble understanding this sentence in light of Figure A.13 which shows an increase in total N gas production from the centre to the northern regions, whereas the sentence seems to imply that the change only occurs for the EF. Can this be written more clearly?

Figure 4. and associated discussion on change in N₂O emissions due to changing spatial distribution of Fertiliser N inputs. I'm sorry, I have had a little bit of trouble understanding this bit and associated assumptions in the calculations for Figure 4. The caption states 'a) The average change in annual N₂O emissions due to the shifting spatial distribution of fertiliser N inputs, calculated using the 1940 spatial distribution of fertilisation scaled to the 2020 total fertilisation quantity, and comparing this to the actual 2020 emissions'. To me this indicates that the assumption has been made that the distribution of fertiliser usage remains the same in 2020 as it was in 1940, just the amount increases. Can the authors clarify how this has been determined? Has the adjustment been made based on 2020 data of usage for each country? If the former then it would seem that the spatial distribution of N inputs is not changing, just increasing, and it is hard to then comprehend the discussion in lines 323 – 327. The methods section mentions access to 2 datasets of fertilizer N inputs. Was this used for this calculation? The conclusion states that the N EFs have increased because of spatial redistribution of fertiliser N inputs. This implies the data in Figure 4 has accounted for changes between and potentially within countries. Further evidence or some clarity in description of the data is needed here to enable these conclusions to be supported.

Figure A14. How is the simulated proportion of N lost from all inputs determined? X-Axis 2020 is missing 0. If the authors use the term EF in the manuscript (i.e. line 322) I wonder if it would be helpful to use this term in the figure caption as well.

Line 352. This line states that 'deposition inputs drive N₂O emissions' but seems to contradict the data presented in Figure 2 which shows fixation as the main source of N₂O. there is some confusion about the N₂O emissions comparing Table 1, and Figures 2 and A11. Can the authors please make clear the reason for differences in the presented data?

I feel the conclusion could benefit from a comment on how well the 'novel coupled soil-atmosphere model' was able to quantify terrestrial N losses and N₂O EFs, as this is listed in the abstract as a focus of the manuscript. Figure 1 of the manuscript also refers to comparison of the novel model with another approach.

Check format for citation of multiple authors

Line 591-592. As the total N input has increased, even if N₂O, NO and N₂ increase with temperature, would leaching necessarily decrease?

As MRPI is defined as the N₂O mixing ratio for the preindustrial troposphere (Table A1), the term MRPI, strat seems confusing – perhaps MRPI, trop should be used for the definition of the mixing ratio in the troposphere.

Line 611 – do the authors mean moles of N₂O in the troposphere and stratosphere?

After line 647. Gaussian and uniform parameters are represented by PG and PU while in Table A1 they are represented by G and U. It would make it clearer for the reader if these were the same.

Line 668. Should this reference be to Table A1 rather than Table 1?

Edit reference to reference in line 678

Line 171, 717, 811-812 , 841, 920 – edit sentences

Figure A4. Is the caption correct? The caption makes reference to bulk density not MAP. This figure should be placed after line 836.

Line 830 What is AI ? is this aridity index? This acronym should be defined.

Figures A.5 and A.6 Reference is made twice to Eq. A.3. and not to Eq A.4, is this an error?

Define W in Eqns A.1 to A.6

Regarding the source of N₂O, there are more papers on laboratory based investigations of the source of N₂O (from nitrification or denitrification). Would these be suitable sources of data to improve the size of the data available?

Figure A.9. please explain what is meant by prior and post observations. The last line of the caption states 'the posterior observations are significantly different to the prior estimates' therefore are they observations or modelled estimates?

Reviewer #3 (Remarks to the Author):

This manuscript by Harris et al. uses a novel coupled soil-atmosphere isotopic modeling approach to examine changes in atmospheric N₂O concentrations and terrestrial N₂O sources since 1850. They show that N₂O emission factors are likely underestimated by the ICPP, and that future increases in terrestrial N₂O emissions may be driven by increased N fertilization by emerging economies with warmer climates. These results paint a comprehensive picture of terrestrial N₂O sources and pinpoint key areas for future mitigation. The manuscript is well written and the results will be useful to a broad community of scientists interested in N₂O sources and climate change. The theory behind the modelling approach appears sound, but I am not qualified to thoroughly evaluate the modelling methods used in this study. I believe these results will be of interest to readers of Nature Communications and have only a few comments for the authors to address:

L62 – 63: The authors could define NO and N₂ in the introduction. While N₂O is the focus of this story, it would be useful to know why we should also care about these other trace gases.

L104: It is not clear to me what an “inversion framework” is. The authors should define this for a broad audience that reads Nature Communications.

L122: I found the transition from discussing soil ¹⁵N to N₂O isotopes a little jarring. It might be useful for the authors to clearly state how looking at both soil ¹⁵N as well as N₂O isotopes can improve models of terrestrial N₂O sources.

L158 – 159: The authors should clarify what they mean by natural vs anthropogenic emissions. Do “anthropogenic emissions” mean the N is derived from anthropogenic sources, the soils are managed, or something else?

L222 – 230: If EFs are underestimated in other modeling frameworks, “particularly at agricultural sites”, wouldn’t that result in them underestimating N₂O from fertilization, not from N deposition? Maybe I am missing something here, but I think the authors could more clearly explain why N deposition accounts for most N₂O emissions in their model, despite fertilization being the largest N input (L65).

L255: It would be clearer if the authors said “N₂O source” rather than just “source” in this topic sentence.

L255 – 270: The authors could more clearly state what the opposing trends in SP and ¹⁵N-N₂O tell us about N₂O sources in this paragraph. It is not clear to me why a decreasing trend in ¹⁵N-N₂O indicates an increase in agricultural emissions, or if agricultural emissions could also explain the trend in SP data.

L271: It would be helpful for the authors to define what an emission factor is somewhere in the introduction.

L274: Figures 3 and A.13 do not seem to support this statement. Figure A.3 might be more appropriate.

L328: I believe that “vary” should be “very”

L397: There is no citation for the data from Africa.

L497: The authors should also discuss why they do not include chemodenitrification in the model.

L519: I believe the authors meant to cite section 4.3, not 4.4.

L674: It is my opinion that the authors should make all data used in this manuscript openly available immediately upon publication.

Reviewer #4 (Remarks to the Author):

This study provides definite progress in estimating global fluxes of N gas emissions especially since Isotone attempts to distinguish between different N₂O production pathways. It provides an expected, but important confirmation, that tier 1 IPCC EFs for N₂O emissions derived using historical fertilizer studies may not extend well across high emission areas and will not consider feedback from a warming climate on soil processes. There is room for improvement in the study as more research becomes available, especially regarding non-linear EFs as influenced by nitrogen use efficiency in agricultural systems. I would suggest that the authors be careful not to discount this relationship and provide more explanation since many research studies and also agricultural producer groups stress the importance of N use efficiency on N₂O emissions (which impacts EF). The manuscript is well written, the methods are acceptable, and the paper tells a compelling story thus I suggest acceptance after the following revisions are considered.

Specific comments:

Line 51: Perhaps be more specific than “redistribution of fertilization”. Are the authors primarily referring to increased fertilizer?

Line 81: Also include N runoff?

Lines 138-139: Do these databases include off-season emissions, which are especially important in cool regions?

Results and discussion

Lines 164-165: If changes in fertilizer application rates for agriculture are considered as part of land-use change then the land-use change should impact N₂O emission factors. Please see my comments for lines 570-573. I do understand that the impacts of fertilizer rate on N use efficiency for cropland are complex depending on soil-plant-crop interactions. Further, it is nice to see that IsoTONE aligns fairly well with inversion estimates in 2020, but I don't think this precludes that land-use change may have a significant influence on N₂O emission factors.

Line 171: Revise to “...land-use change and cultivation...”

Lines 186-189: Perhaps the graph in figure A.10 is not drawn correctly? In panel e) the proportion of N₂O emissions from nitrification and denitrification looks to be considerably lower than 60 and 40%, respectively, by 2020.

Lines 204-206: In alignment, with the idea that farmers will apply greater quantities of N in the future in areas where crop growth becomes more favourable see section “3.3.2. Fixed fertilizer” and Fig.8 in this paper <https://doi.org/10.1016/j.scitotenv.2020.138845>. Greater N application will also be needed, especially in cool humid regions, to compensate for increased N losses under a warming climate.

Lines 226-226: Very interesting. I had also thought N₂O from fertilization would be the highest.

Lines 226-227: A large number of field studies also don't account for off-season emissions in cool climates (especially the large emissions associated with spring thaw events), due to difficulty in measuring these emissions using chambers.

Lines 276-277: perhaps modify EF to “N₂O EF”.

Lines 310-314: A more recent reference: <https://doi.org/10.1016/j.cosust.2020.08.002> Also, as mentioned above, most studies do not include off-season emissions which can be high in temperate climates (<https://www.nature.com/articles/ngeo2907>).

Figure 4: Considering that temperature is increasing at 2x the global rate in Canada I would have expected a temperature-dependence increase in N₂O emissions across Canada from 1940 to 2020.

Methods

Line 380: The description of the Python function could be moved to line 380 since this is when skipy is first mentioned.

Line 385: Was global mean WFPS estimated over time? Annually? There was only mention of annual mean precipitation above in line 369. Is annual WFPS used as another general aridity index? The seasonal temporal trend of WFPS over time is important for driving N₂O emissions. Also, the range of WFPS which impacts N₂O emissions is soil dependent since porosities differ (please see my comment below). WFPS should be defined in line 385 here where it is first mentioned.

Line 408: A tropospheric

Between lines 535 and 536: Why is the fraction of NH₃ always <0.05? I doubt this would be the case for some large agricultural regions when surface-applied urea is dominant.

Regarding this statement: “Mean WFPS will not describe all variability in gas partitioning, given the highly variable and non-linear nature of N gas emissions, however, it provides the best available estimate based on the current status of experimental and modelling research.” Some models such as DNDC estimate soil oxygen content to drive N₂O emissions. Could you please explain how WFPS is applied as a driver for N₂O in your model since the impact should be soil specific with optimal N₂O production generally occurring a little above the upper water holding limit (field capacity)? This may be at 40% WFPS for a sandy loam but 85% for a Clay.

Lines 570 to 573 There may be publications which suggest EF is not strongly dependent on fertilisation rate, however, many other publications suggest that increases in N₂O emissions are non-linear as fertilizer rate increases and likewise there are lower EFs for well-managed systems. This is thought to be common knowledge amongst some scientists. Perhaps the impact of fertilizers on the global scale EFs is confounded by other drivers of N₂O which are not well characterized. Please see this policy brief from an OECD funded meeting which states “There is a clear non-linear relationship between fertilizer application and N₂O emissions, with emissions increasing exponentially when application rates exceed plant N requirements.” [https://mopga.imk-ifu.kit.edu/sites/mopga/files/inline-files/Policy Brief - Nr-Workshop Garmisch final.pdf](https://mopga.imk-ifu.kit.edu/sites/mopga/files/inline-files/Policy%20Brief%20-%20Nr-Workshop%20Garmisch%20final.pdf). This has particularly been a problem in China where fertilizer has been greatly over applied (<https://doi.org/10.1016/j.eja.2021.126409>). To be clear, I think it’s OK to use linear EFs in your global-scale study but it would be preferable to note that EFs may be strongly dependent on fertilizer rates and more research is needed to resolve the influence.

Line 668 “show” should be “shown”

Line 817-818: Was the impact of N availability low due to the highly variable soil N since it “showed no correlation between the point and gridded datasets” as indicated from lines 716-717? The relationship between EF and N availability is important, however, to capture the relationship a robust measurement or process-based modelling study is needed. I would suggest that the authors be careful not to discount this relationship as many research studies and also agricultural producer groups stress the importance of N use efficiency in cropping systems on N₂O emissions.

Figure A.5 Likely we can expect a smaller N₂O/(N₂O + N₂) ratio in fine-textured soils (denitrification is more apt to go to N₂ in clay soils). Have the authors considered developing equations with coarse, medium and fine-textured soils? Again, as mentioned above, WFPS is not a very good measure of N gas emissions across soil types.

Figure A.10 Please revise the caption as b) map of N lost to nitrification is not mentioned. Also, why does the proportion of N₂O from both denitrification and nitrification decline over time? What is the other source of N₂O emissions?

Review response for manuscript NCOMMS-22-00281-T

Eliza Harris et al.

We thank the reviewers and the editor for their careful and thorough consideration of our manuscript. The comments were positive and constructive, and we present a response to each point below. The comments of reviewer #1 are presented first followed by the comments of reviewer #2 and then reviewer #3; for each reviewer all points are addressed in the order in which they appear in the reviews. Line and page numbers refer to the original submission, not the revised submission. All changes made to the manuscript are highlighted in this document and in the revised manuscript.

Reviewer #1 comments

- 1. On a first read of the manuscript, it was quite difficult to keep track of all the figures and data, and the presentation of the supplementary figures did not seem to follow a logical (chronological) order in the manuscript. For example, Figure A.12 is the 1st figure referenced in the manuscript (line 146). I wonder if it is possible to present the data in a manner that does not require continuous reference to the supplementary figures for the reader to more readily comprehend the information.**

We hope that the significant changes made in response to other comments from the reviewers, including changes to some figures and clarification of many points, have made the main manuscript more readable as a standalone document. We have also changed the text in section 2.1 (L142-177) so that the references to supplementary figures are less prominent and follow the primary results presented in the main manuscript.

The order of the supplementary material follows a topical structure that relates to the order of information in the manuscript, eg. input data, methods, specific results, other figures and tables. Although this means that the figure cross references are not in the same order as in the main manuscript, we believe this is important to make the supplementary material readable. The extensive supplementary material is necessary to provide detailed methods and results, while keeping the main manuscript readable and accessible to a wider audience.

- 2. I was also a little unclear on the focus of the manuscript. The title and much of the discussion is about the predicted emissions and reasons for any changes from 1850 to 2020. So is the focus about the ability of the model to predict N₂O emissions or about N₂O emissions predicted by the model? This might affect the arrangement of the presented figures.**

The two points made by the reviewer are together the main foci of this manuscript: The new and novel modelling technique, and the results it can predict. We have changed the title to clarify the temporal scale, focussing on 1850-2020:

Warming and redistribution of nitrogen inputs drive an increase in mean terrestrial nitrous oxide emission factor over the past two centuries

We have furthermore changed the order of the 'Key points' to address first the approach, and then the results.

The main manuscript follows this order: First the model description and optimization as well as the comparison to other studies is presented, followed by more detailed results of the approach. We hope that the change in the title and the key points will improve the clarity of the take-home message.

- 3. Please clarify terminology EF. Is this emission factors or emission factor? Please check throughout the manuscript.**

EF stands for Emission Factor, so the plural is EFs; this is now consistent throughout the manuscript.

- 4. Line 146. Figure A.12 is referenced for terrestrial N₂O emissions and comment is made on emissions in 1860 and 2010. However Figure A.12 only shows the 2020 data. Also in line 157, reference to the year 2020 should be made.**

1860 and 2010 results are reported here to allow direct comparison to results reported from the N₂O Model Intercomparison Project [31]. However, in the figure we wished to show the most up-to-date results (2020). This is now clarified in the text:

Total terrestrial N₂O emissions for 1860, 2010 and 2020 were 5.3 ± 0.4 , 12.6 ± 1.2 and 13.9 ± 1.4 Tg N₂O-N a⁻¹ respectively (2020 results shown in Figure A12), showing good agreement with 1860 and 2010 estimates of 6.3 ± 1.1 and 10 ± 2.2 Tg N₂O-N a⁻¹ from the N₂O Model Intercomparison Project [31].

At L157 a reference to 2020 was also added.

- 5. Figure A.12 mentions anthropogenic emissions as being the increase above those in 1800 whereas elsewhere in the manuscript 1850 is referred to as the baseline (e.g. line 223, Figure A.11). Can the authors clarify what the baseline year is in the modelling of anthropogenic emissions?**

The model runs started in 1800 but no datasets included anthropogenic N inputs or emissions until 1850, therefore the baseline was 1850; we apologise for the confusion, and have clarified this in the text.

L558: As none of the N input datasets included significant anthropogenic inputs before 1850, 1850 is taken as the preindustrial 'baseline' throughout this study.

The caption of Figure A12 was also changed.

- 6. Table 1. Please make clear the meaning of 'Growth rate is the 10-year mean growth rate centred on the year of interest.' Does this mean the 10 year average for 1850 is the average from 1845-1855 (as 1850 is the centre of that) or from 1840 to 1850?**

The 10-year average for 1850 is the average from 1845-1855. This is now clarified in the table caption:

Growth rate is the 10-year mean growth rate centred on the year of interest, ie. the 10-year average for 2000 is the average from 1995-2005.

- 7. Table 1 should be placed before Figure 1 according to the order of citing in the manuscript**

This has been corrected.

- 8. Line 144-145. Reference is made to the model outputs agreeing with previous results. Are the previous results from the literature and if so, please add citations. If not from the literature, can these previous results be shown?**

Here we were referring to the comparisons cited in lines 146-155; this is now clarified:

Total terrestrial N₂O emissions from the optimised model (Table 1) agree well with previous results, providing confidence in the novel isotopic basis of the model, for example: Total terrestrial N₂O emissions...

9. **Table A1. The term PD is being used for present day (i.e. in description of ‘Ratio of N₂O lifetime...’ in unitless and years descriptions) and probability distribution (last column of table). Please modify one or other to make this clear.**

PD is more commonly used to refer to the present day, therefore we have changed the abbreviation for probability distribution to ‘PDF’ (probability density function) in the caption and the table.

10. **Terminology is not defined. For example, Table A1. SPOcean is defined as Mean SP for the ocean source but SP should be defined also. Pre-industrial (PI) should be defined in the 1st row of data (MRPI) not the 5th row.**

We have fixed the definition of PI. We have changed ‘SP’ to the more accurate term ‘ $\delta^{15}\text{N}^{\text{SP}}$ ’, throughout the paper, and added a definition of the isotopic terms at the first instance:

N₂O bulk 15-N isotopic composition (\$\delta^{15}\text{N}^{\text{bulk}}\$ ), hereafter abbreviated as \$\delta^{15}\text{N}\$, and N₂O 15-N isotopic site preference, hereafter referred to as \$\delta^{15}\text{N}^{\text{SP}}\$. See [38] for definitions and review.

11. **Line 884 refers to Section A.3 and is in Section A.3, suggest delete**

This has been corrected.

12. **How is the Isotone model better than the CAMS inversion model?**

The IsoTONE model is not ‘better’ than the CAMS model; it has a totally different basis and goals to most existing models, which is the focus of this paper. Comparison of IsoTONE and CAMS provides a validation of both modelling methods: IsoTONE agrees well with CAMS, as stated in this paper, providing confidence in the the results from both models. IsoTONE can be easily applied at different spatial scales, is fast to run on a regular laptop, and can provide information about emission factors and processes in a different manner to CAMS.

We have added a summary of the differences between IsoTONE and CAMS to the Conclusion section:

Compared to CAMS, IsoTONE is able to harness isotopic information to understand spatial variability in emission factors and production and loss processes; moreover, the simplified approach means IsoTONE has low computational requirements and can be used to explore questions requiring many simulations.

13. **Figure A10. Chart e). It is not clear how the proportion of N₂O emissions (%) from nitrification and denitrification both reduce from 1800s. In 1800’s this added makes 100% (40+60) but in 2020 the total is only 90%. What pathway is producing the remaining 10% of N₂O emissions? Caption states percentage for charts d) and e) but y-axis states proportion. Please use consistent terminology.**

The remaining emissions are direct emissions from EDGAR for categories 1A1, 1A3b, 2B and 6, which is now clarified in the figure caption. There are no direct anthropogenic emissions in 1800, thus the contributions of nitrification and denitrification add up to 100%. The terminology has also been corrected. We additionally fixed an error in the numbering of the subplots in this figure. The caption now reads:

N losses to nitrification and denitrification. a, b) Global map of N losses in 2020 on a \$0.5 \times 0.5^\circ\$ degree grid via denitrification (a) and nitrification (b). c) Total N losses to nitrification and denitrification per year from 1800 to 2020 (left axis) compared to total terrestrial N inputs (right axis). d) Proportion of total terrestrial N inputs lost to nitrification and denitrification from 1800 to 2020. e) Proportion of terrestrial N₂O emissions contributed by nitrification and denitrification from 1800 to 2020 (remaining N₂O emissions are from EDGAR categories 1A1, 1A3b, 2B and 6, see Section 4.2.2).

14. **Fert_EF_red (Table A1) is a little confusing until reading lines 559-581. As this is a factor, I wonder if it should be written as EF reduction for fertiliser emissions factor (then matches the fractionation expression factor). I also think it would be helpful to point the reader to the section where this (and other terms in the Table) are defined.**

We have added a reference to the appropriate sections so that readers can find the definition of fert_EF_red easily.

15. **Comments regarding Figure 2:**

Figure 2 2nd panel and Figure A.11 1st panel seem to be presenting the same data but the values are different on the y-axis although the caption seems to state the same assumptions. Can the authors make clear what the difference is between these 2 figures? Is it because the A.11 figure is based off estimates using 15N and Figure 2 uses non-labelled findings?

The data in these figure panels are identical. Anthropogenic N₂O emissions from different input categories is a key result, and thus it is included as a subpanel of Figure 2 in the main article. It is also complementary to the isotopic values presented in Figure A11. The results previously looked different because the natural background was separated in Figure 2 and the x scales were different. The two panels are now presented more similarly so it can be clearly seen that the panels are the same. Moreover, additional information is included in Figure A11 (natural fixation and deposition emissions, pre-1900 data) and referenced in the Figure 2 caption.

Figure 2 2nd panel and Table 1 data do not seem to match. Even though Table 1 presents the 10 year data, the values provided for fixation N increase from 0.2 to 1.8 whereas Figure 2 shows a value that looks stable at around 4. And the total anthropogenic soil emissions in Table 1 increase from 1 to 7 but in Figure 2 appear to increase from 5 to 10.

This was due to the distinction between natural and anthropogenic sources, which was confusing as the total emissions were shown in the previous version of Figure 2. The relationship between is clearer now that the emissions are split into anthropogenic and natural categories.

Figure 2. The scale starts at 1900 rather than prior to 1850 as shown in Figure A.11 and in Table 1.

This is to 'zoom in' on the period of significant growth from 1900 onwards. A cross reference to the longer time series of emissions data in Figure A11 is now included in the Figure 2 caption. Moreover, a cross-reference to Figure 2 is included in the Figure A11 caption.

Line 210 The authors state the anthropogenic N2O flux in 2020 is 7.1 Tg per annum, but Figure 2 indicates a total of around 10 Tg per annum.

The soil source is 7.1 Tg per annum, whereas the blue curve (total emissions) also includes the non-soil source to give a total of 8.8 Tg per annum. This is now explicitly stated in the caption.

Figure 2 panel 3 has more lines of data (8) than provided in the legend (6) and the 3 year mean and 10 year mean colours are not visible in the chart.

The colours of the lines follow the fertilisation-fixation-deposition-total colour scheme and the pale/dark lines show the 3 and 10 year means; this is now clarified in the caption.

Line 237 - does the Figure 2 panel 3 y-axis scale reach 0.3? It is difficult to see

0.3 has been added as a tick mark on this axis.

Sometimes a-1 a-1 is used and sometimes a-2, should only 1 of these be used?

This has been changed to a⁻¹ a⁻¹ throughout, which give greater clarity than a⁻², as this refers to a change in a rate.

The updated versions of Figure 2 and A11 and captions are included:

Figure 2: Temporal evolution of N inputs and N₂O emissions, and growth rates of N₂O emissions and N₂O mixing ratio. The top panel shows annual inputs for fertilisation, deposition, fixation, and total N used in the model (see Section 4.1 for data sources). The second panel shows the anthropogenic flux broken down into N input categories of fertilisation, deposition and fixation, estimated by assuming that all increases in N₂O emissions for all input categories (fixation, fertilisation, deposition) after 1850 are due to anthropogenic influences. Total N₂O emissions, including non-soil N₂O emissions, are also shown (blue line). The shaded areas indicate the 1σ uncertainty. N₂O emission data prior to 1900 as well as the breakdown of natural emissions driven by deposition and fixation are shown in Figure A.11. The third panel shows the growth rate of N₂O emissions from each input category calculated over 3 and 10 year windows (pale and dark lines respectively, using colours indicated for fertilisation, fixation and deposition). The bottom panel shows the growth rate in modelled N₂O tropospheric mixing ratio (left axis; purple solid line) as well as the modelled change in mixing ratio growth rate (right axis; blue dotted line).

Figure 11: Anthropogenic and natural terrestrial N_2O emissions and isotopic composition. The top panel shows the anthropogenic flux broken down into N input categories of fertilisation, deposition and fixation, estimated by assuming that all increases in N_2O emissions for all input categories (fixation, fertilisation, deposition) after 1850 are due to anthropogenic influences. Total N_2O emissions, including non-soil N_2O emissions, are also shown (blue line). The bars at the left indicate the breakdown of natural emissions driven by deposition and fixation. The bottom panels show the $\delta^{15}N$ and $\delta^{15}N^{SP}$ for natural and anthropogenic emissions and for total terrestrial emissions. N inputs with time as well as growth rate of N_2O emissions for each input category are shown in Figure 2. The shaded areas indicate the 1σ uncertainty.

16. **Figure 3 should be moved further down the manuscript to after line 274.**

The figure has been moved to the end of the relevant paragraph. The figure positions will also be carefully checked at the typesetting stage.

17. **Line 286-287. I have a little trouble understanding this sentence in light of Figure A.13 which shows an increase in total N gas production from the centre to the northern regions, whereas the sentence seems to imply that the change only occurs for the EF. Can this be written more clearly?**

Figure A.13 shows the proportion of N lost to gas production, and not the total gas production, whereas Figure A.12 shows the total gas production. The sentence is now written more clearly and figure A.13 has been edited to emphasize that the proportion lost to gas production. The sentence now reads:

Geoclimate-driven variability in emission factors can be clearly seen in regions underrepresented in EF compilations, such as Australia, which shows a clear gradient from low to high N₂O EFs between the dry centre and wetter coastal and northern regions, but very little gradient in total N gas production.

18. **Figure 4. and associated discussion on change in N₂O emissions due to changing spatial distribution of Fertiliser N inputs. I'm sorry, I have had a little bit of trouble understanding this bit and associated assumptions in the calculations for Figure 4. The caption states 'a) The average change in annual N₂O emissions due to the shifting spatial distribution of fertiliser N inputs, calculated using the 1940 spatial distribution of fertilisation scaled to the 2020 total fertilisation quantity, and comparing this to the actual 2020 emissions'. To me this indicates that the assumption has been made that the distribution of fertiliser usage remains the same in 2020 as it was in 1940, just the amount increases. Can the authors clarify how this has been determined? Has the adjustment been made based on 2020 data of usage for each country? If the former then it would seem that the spatial distribution of N inputs is not changing, just increasing, and it is hard to then comprehend the discussion in lines 323 – 327. The methods section mentions access to 2 datasets of fertilizer N inputs. Was this used for this calculation? The conclusion states that the N EFs have increased because of spatial redistribution of fertiliser N inputs. This implies the data in Figure 4 has accounted for changes between and potentially within countries. Further evidence or some clarity in description of the data is needed here to enable these conclusions to be supported.**

We apologise for the lack of clarity here. The impact of changing spatial distribution was estimated by using comparing actual modelled 2020 emissions, calculated using a full gridded fertiliser input dataset (Land Use Harmonization Database), and comparing this to a baseline representing no change in fertiliser input distribution since 1940. The latter was calculated, as you mention, by taking the 1940 fertiliser inputs and scaling up to the 2020 totals. The full modelled 2020 distribution of inputs and emissions was used in all other analyses throughout the paper; the 1940-scaled input and emissions only as a baseline for this analysis. The actual 2020 distribution of fertiliser N inputs showed much more N in regions with relatively high EFs compared to the 1940-scaled inputs and thus led to a significant amount of emissions that were not due to additional fertiliser, but due to this shift from lower to higher N₂O-EF regions. We have tried to clarify this in the figure caption and in the text.

Added to the text at L324: This has two major causes: Changing spatial distribution of N inputs and climate warming feedback (Figure 4). The impact of changing spatial distribution of N inputs on annual N₂O emissions and EFs was estimated by calculating a baseline, using the 1940 spatial distribution of fertilisation scaled to the 2020 total fertilisation quantity, and comparing this to the modelled 2020 maps of emissions and EFs. The shift in fertilisation

Figure 4 caption: The average change in annual N₂O emissions due to the shifting spatial distribution of fertiliser N inputs, calculated using the 1940 spatial distribution of fertilisation scaled to the 2020 total fertilisation quantity as a baseline, and comparing this to the actual modelled 2020 emissions.

19. **Figure A14. How is the simulated proportion of N lost from all inputs determined? X-Axis 2020 is missing 0. If the authors use the term EF in the manuscript (i.e. line 322) I wonder if it would be helpful to use this term in the figure caption as well.**

All input and output categories are tracked in IsoTONE, making it a basic property of the model code to determine the proportions of N lost from each input category. The x-axis has been fixed, and the term EF has been added to the caption:

The simulated proportion of N from different input categories (fertilisation, deposition, fixation) that is lost via leaching, NH₃ volatilization, and gas production (EFs for N₂, N₂O, NO) in soils from 1940-2020.

20. **Line 352. This line states that ‘deposition inputs drive N₂O emissions’ but seems to contradict the data presented in Figure 2 which shows fixation as the main source of N₂O. there is some confusion about the N₂O emissions comparing Table 1, and Figures 2 and A11. Can the authors please make clear the reason for differences in the presented data?**

The differences between Table 1, Figure A.11 and 2 have been addressed and clarified in response to comment 15. L352 is referring to anthropogenic emissions, and this is now clarified in the text; in addition, the breakdown of natural emissions into fixation and deposition terms has been added to Table 1. L352 now reads:

The model results show that fixation N inputs drive the majority of natural N₂O emissions, but deposition N inputs accord for the majority of anthropogenic emissions.

21. **I feel the conclusion could benefit from a comment on how well the ‘novel coupled soil-atmosphere model’ was able to quantify terrestrial N losses and N₂O EFs, as this is listed in the abstract as a focus of the manuscript. Figure 1 of the manuscript also refers to comparison of the novel model with another approach.**

We have added this to the conclusion, also following from Comment 12 of reviewer 1:

We developed and applied a coupled soil-atmosphere nitrogen isotope model framework - ‘IsoTONE’ - using soil δ¹⁵N as an emission proxy to understand the processes underlying spatiotemporal dynamics of global N₂O emissions. This novel model set up used isotopic composition to trace different N loss and N₂O production processes, with a Markov Chain Monte Carlo approach implemented to constrain the model using tropospheric time series of N₂O isotopic composition. Results from IsoTONE agree well with the CAMS inversion model [28], providing confidence in both methodologies. Compared to CAMS, IsoTONE is able to harness isotopic information to understand spatial variability in emission factors and production and loss processes; moreover, the simplified approach means IsoTONE has low computational requirements and can be used to explore questions requiring many simulations.

22. **Check format for citation of multiple authors**

This will be fixed when the correct bibtex citation style is used in typesetting.

23. **Line 591-592. As the total N input has increased, even if N₂O, NO and N₂ increase with temperature, would leaching necessarily decrease?**

Yes, if temperature have driven higher gas production, we expect reduced leaching to maintain mass balance. This is stated at L592: ‘N losses through leaching were consequently reduced to maintain N mass balance.’

A number of other processes could also impact leaching, particularly under future climate conditions. For example, increased nitrification in a warming climate may increase total nitrogen cycling, thus increasing the total amount of leached N though this may not affect the proportion of losses due to leaching. However, plant N uptake may also increase in a warm climate with elevated CO₂, reduces losses via other processes. Changes in rainfall frequency and intensity will also impact leaching. Data to constrain these processes is starting to become available, and incorporation of these processes will be key to allow IsoTONE to predict N cycling in a future climate.

24. **As MRPI is defined as the N₂O mixing ratio for the preindustrial troposphere (Table A1), the term MRPI_{strat} seems confusing – perhaps MRPI_{trop} should be used for the definition of the mixing ratio in the troposphere.**

This is a good suggestion but we fear it would make Eq. 7 much more difficult to read. We have therefore clearly defined these two terms when they first appear: where MR_{PI} and MR_{PI, strat} are the N₂O mixing ratios in the preindustrial troposphere and stratosphere respectively.

25. **Line 611 – do the authors mean moles of N₂O in the troposphere and stratosphere?**

No, the total moles of air in the troposphere and stratosphere. This is now clarified: ...moles_{trop} and moles_{strat} are the moles of air in the troposphere and the stratosphere (1.5 and 0.27 × 10²⁰ moles respectively).

26. **After line 647. Gaussian and uniform parameters are represented by PG and PU while in Table A1 they are represented by G and U. It would make it clearer for the reader if these were the same.**

P represents the parameter value in the section after line 647 whereas G and U are used to represent the distribution, like in Table A1. We have edited the sentence at L647 for clarity:

Within each iteration i of the MCMC, parameters following a Gaussian uncertainty distribution (see Table A.1), were varied according to:

$$P_{i+1,G} = P_{i,G} + 1\sigma \text{ uncertainty} \times \text{stepsize} \times r_{\text{unif}} \quad (1)$$

where \$P_{i,G}\$ is the value of the Gaussian parameter in iteration \$i\$ and r_{unif} is a uniformly distributed random number between -1 and 1. Uniform parameters were varied according to:

$$P_{i+1,U} = P_{i,U} + \frac{R_{\text{max}} - R_{\text{min}}}{4} \times \text{stepsize} \times r_{\text{unif}} \quad (2)$$

where \$P_{i,U}\$ is the value of the uniformly-distributed parameter in iteration \$i\$, and...

27. **Line 668. Should this reference be to Table A1 rather than Table 1?**

Yes, this has been corrected.

28. **Edit reference to reference in line 678**

This has been corrected.

29. **Line 171, 717, 811-812 , 841, 920 - edit sentences**

All sentences have been corrected.

30. **Figure A4. Is the caption correct? The caption makes reference to bulk density not MAP. This figure should be placed after line 836.**

Yes, this has been corrected.

31. **Line 830 What is AI ? is this aridity index? This acronym should be defined.**

Yes, this is aridity index, which is now defined the first time this acronym is used.

32. **Figures A.5 and A.6 Reference is made twice to Eq. A.3. and not to Eq A.4, is this an error?**

Equation 6 was incorrectly labelled in the latex source which led to this error; this has been corrected in both figures.

33. **Define W in Eqns A.1 to A.6**

This has been changed to WFPS. A definition of the acronym WFPS was missing from the manuscript, and has been added at the first instance at L385: water-filled pore space (WFPS)

34. **Regarding the source of N₂O, there are more papers on laboratory based investigations of the source of N₂O (from nitrification or denitrification). Would these be suitable sources of data to improve the size of the data available?**

The cited papers represented the available data at the time of writing as far as we were able to find. The increasing use of techniques such as isotope spectroscopy mean that in the coming years, a great deal more data should be available to improve this parameterisation in future versions of IsoTONE.

35. **Figure A.9. please explain what is meant by prior and post observations. The last line of the caption states ‘the posterior observations are significantly different to the prior estimates’ therefore are they observations or modelled estimates?**

These are the estimates of EFs, based on observations before and after optimization using a Bayesian approach with the MCMC algorithm. This has been clarified:

both the prior and posterior estimates are shown with 1σ error bars, as the posterior estimates following the MCMC

Reviewer #2 comments

1. **L62 - 63: The authors could define NO and N₂ in the introduction. While N₂O is the focus of this story, it would be useful to know why we should also care about these other trace gases.**

A brief overview and citation has been added: The primary global source of N₂O is production during N cycling by microbiota in soils. Soil N cycling also releases NO and N₂, which directly impact tropospheric ozone production, climate, and soil N content and loss pathways [10, 23].

2. **L104: It is not clear to me what an “inversion framework” is. The authors should define this for a broad audience that reads Nature Communications.**

We have edited this sentence to clarify that inversion approaches aim to iteratively improve parameterisations: ...which makes it difficult to iteratively constrain and optimize model parameters with observations using typical inversion frameworks and likelihood approaches, and complicates investigations of global or long-term emissions.

3. **L122: I found the transition from discussing soil 15N to N₂O isotopes a little jarring. It might be useful for the authors to clearly state how looking at both soil 15N as well as N₂O isotopes can improve models of terrestrial N₂O sources.**

We have improved this transition by adding a link between these two data sources: δ¹⁵N_{soil} can be a proxy for soil N loss pathways, including N₂O production. This approach can be extended to include records of atmospheric N₂O isotopic composition, which reflects N₂O production pathways [32, 19, 18], and can be used to validate results from studies of δ¹⁵N_{soil}.

4. **L158 - 159: The authors should clarify what they mean by natural vs anthropogenic emissions. Do ‘anthropogenic emissions’ mean the is N derived from anthropogenic sources, the soils are managed, or something else?**

This is now defined in the text at L158: The spatial distribution of emissions from fertilisation, deposition and fixation for natural and anthropogenic soils for 2020 is shown in Figure A.12, with ‘anthropogenic emissions’ referring to all emissions above the preindustrial baseline, thus accounting for both direct and indirect anthropogenic N₂O sources.

5. **L222 – 230: If EFs are underestimated in other modeling frameworks, ‘particularly at agricultural sites’, wouldn’t that result in them underestimating N₂O from fertilization, not from N deposition? Maybe I am missing something here, but I think the authors could more clearly explain why N deposition accounts for most N₂O emissions in their model, despite fertilization being the largest N input (L65).**

This was unclear - we mean that measurements at agricultural sites will lead to low estimates of EFs, and these will consequently lead to low estimates of deposition-driven emissions when applied to other sites. The text is now written more clearly:

EFs will be particularly underestimated from field measurements made at agricultural sites, where a significant proportion of N is removed through harvest or immobilization as described by fert_EF_red, thus leading to low EFs. These combined effects will lead to underestimation of deposition emissions when measured EFs are applied at non-agricultural sites in bottom up-N₂O emission frameworks.

6. **L255: It would be clearer if the authors said ‘N₂O source’ rather than just ‘source’ in this topic sentence.**

This has been changed.

7. **L255 – 270: The authors could more clearly state what the opposing trends in SP and 15N-N₂O tell us about N₂O sources in this paragraph. It is not clear to me why a decreasing trend in 15N-N₂O indicates an increase in agricultural emissions, or if agricultural emissions could also explain the trend in SP data.**

We have improved the clarity and structure of this paragraph:

The isotopic composition of the total anthropogenic N₂O source reflects changing emission processes (Figure A.11). During the growth rate acceleration of 1945-1980, $\delta^{15}\text{N}$ and $\delta^{15}\text{N}^{\text{SP}}$ of the anthropogenic source were relatively constant, however after 1980 they changed more rapidly. These fluctuations were also observed by [25], who used a two-box model to interpret N₂O source isotopic composition from firm air data, and found similar but more uncertain values for anthropogenic source isotopic composition. $\delta^{15}\text{N}$ of the anthropogenic source is higher than the mean soil source (Table 1) and shows a strong decreasing trend, which may indicate an increasing dominance of agricultural emissions with low $\delta^{15}\text{N}$ -N₂O after 1980 [25]. An increasing proportion of agricultural emissions could also explain the trend in anthropogenic source $\delta^{15}\text{N}^{\text{SP}}$, which approaches the estimated agricultural mean of $7.2 \pm 3.8\text{‰}$ [38]. Other explanations for changing source isotopic composition include changes in the extent of pathways such as consumption via N₂O reduction and production via nitrification and denitrification. Variability in $\delta^{15}\text{N}^{\text{SP}}$ and $\delta^{15}\text{N}$ are in opposite directions, and thus unlikely to be caused by changes in the extent of N₂O reduction to N₂, which would increase both the $\delta^{15}\text{N}^{\text{SP}}$ and $\delta^{15}\text{N}$ of remaining N₂O [39]. Furthermore, our results suggest a 1% decrease in nitrification contribution to global N₂O emissions and thus very little change in the nitrification:denitrification ratio, thus this could not account for the observed increase of $\sim 1\text{‰}$ in $\delta^{15}\text{N}^{\text{SP}}$ of anthropogenic N₂O based on $\delta^{15}\text{N}^{\text{SP}}$ endmembers of 0 and 30‰ for N₂O from denitrification and nitrification.

8. **L271: It would be helpful for the authors to define what an emission factor is somewhere in the introduction.**

We have added the following text at L84 to define the term ‘emission factor’:

The proportion of N inputs released as particular N gases can be described with the emission factor (EF); for example, an EF for N₂O of 2% means that 2% of annual N inputs are released as N₂O. On the global scale, the impacts of climate change on N-gas production processes are poorly known: Warming is generally expected to enhance microbial activity and increase N-gas EFs, however interactions between factors such as N availability, plant growth, and precipitation changes are poorly constrained.

9. **L274: Figures 3 and A.13 do not seem to support this statement. Figure A.3 might be more appropriate.**

Figures 3 and A.13 show the modelled variability in N gas and N₂O proportional emissions and therefore strongly relate to this statement; we have also added a reference to Figure 3 as suggested.

10. **L328: I believe that “vary” should be “very”**

This has been corrected.

11. **L397: There is no citation for the data from Africa.**

This data is not yet published but we have included a citation to a related paper, and updated this section to clearly distinguish published and unpublished datasets:

Geographical coverage was improved by adding a further 748 samples [12, 8, 37, 21], including unpublished data for 112 samples from Australia (BASE database; [7]) and 392 from Africa (experimental sites and set up described in [4, 3, 5, 6, 17, 1, 2]) - regions which were underrepresented in the Craine database.

12. **L497: The authors should also discuss why they do not include chemodenitrification in the model.**

We have added a reference to both co- and chemodenitrification at this point:

Similarly, co- and chemodenitrification are poorly constrained pathways, both in terms of drivers and isotopic composition [36, 13, 18]. Fungal denitrification and co- and chemodenitrification will be included in a future version of this model as data to drive parameterisations become available.

13. **L519: I believe the authors meant to cite section 4.3, not 4.4.**

Correct; this has been fixed.

14. **L674: It is my opinion that the authors should make all data used in this manuscript openly available immediately upon publication.**

The data will be released along with the publication, both as a standalone dataset on the Pangaea database, as well as part of the model code repository.

Reviewer #3 comments

1. **Line 51: Perhaps be more specific than “redistribution of fertilization”. Are the authors primarily referring to increased fertilizer?**

Here we are referring to changes in the spatial distribution of fertilizer N input, which has occurred alongside increases in overall fertilizer N inputs. This is now clarified:

Climate change and changes in the spatial distribution of fertilisation N inputs have driven...

2. **Line 81: Also include N runoff?**

We have now mentioned runoff coupled to leaching:

N is lost from terrestrial ecosystems through several major pathways: Microbial and abiotic *N gas production* in soils; *runoff and leaching of N species*; and ammonia *volatilization*.

3. **Lines 138-139: Do these databases include off-season emissions, which are especially important in cool regions?**

Unfortunately off-season data is sparse, which is a clear problem in estimating annual emission factors. We have clarified this point at L893 when the EF database is discussed in detail:

Moreover, measured EFs are often based only on growing season emissions, which can lead to a strong underestimation of annual emissions that could be particularly important in cold regions [35, 11, 16].

4. **Lines 164-165: If changes in fertilizer application rates for agriculture are considered as part of land-use change then the land-use change should impact N₂O emission factors. Please see my comments for lines 570-573. I do understand that the impacts of fertilizer rate on N use efficiency for cropland are complex depending on soil-plant-crop interactions. Further, it is nice to see that IsoTONE aligns fairly well with inversion estimates in 2020, but I don't think this precludes that land-use change may have a significant influence on N₂O emission factors.**

Here we refer to the assumption that, on a coarse grid scale at an annual timescale, changes in land use have a minor impact on N cycling and N₂O emissions. This does not include fertiliser inputs, which are explicitly accounted for as inputs, and clearly impact N₂O emissions. We have clarified in the text:

The IsoTONE framework assumes that land use changes - aside from fertiliser use and other N inputs, which are explicitly provided to the model - have had a minor impact on N₂O emission factors at an annual timescale, compared to the major impact of pre-existing variability in emission factors driven by climate and soil parameters. This assumption is supported by the good agreement between CAMS inversion results and IsoTONE emission estimates.

5. **Line 171: Revise to "...land-use change and cultivation..."**

This has been corrected

6. **Lines 186-189: Perhaps the graph in figure A.10 is not drawn correctly? In panel e) the proportion of N₂O emissions from nitrification and denitrification looks to be considerably lower than 60 and 40%, respectively, by 2020.**

This was unclear and has been addressed under comment 13 from Reviewer 1.

7. **Lines 204-206: In alignment, with the idea that farmers will apply greater quantities of N in the future in areas where crop growth becomes more favourable see section "3.3.2. Fixed fertilizer" and Fig.8 in this paper <https://doi.org/10.1016/j.scitotenv.2020.138845>. Greater N application will also be needed, especially in cool humid regions, to compensate for increased N losses under a warming climate.**

This is a really good point and has been mentioned at L204:

Mean global fertiliser N incorporation could be increasing as plant growth and thus N use is enhanced in response to increasing atmospheric CO₂. However, water and nutrient limitations will also play a role in regulating plant growth [30], and fertiliser use may moreover increase in response to higher N use [27]. These potential effects are not currently captured in IsoTONE and should be a focus of future model studies.

8. **Lines 226: Very interesting. I had also thought N₂O from fertilization would be the highest.**

Yes, this was an interesting result.

9. **Lines 226-227: A large number of field studies also don't account for off-season emissions in cool climates (especially the large emissions associated with spring thaw events), due to difficulty in measuring these emissions using chambers.**

This is an important point and has been added:

Moreover, our results suggest that emission factors could be significantly underestimated from field measurements (see Sections 2.1 and 2.3), likely due to the extremely dynamic nature of N₂O emissions, which are not adequately captured with sparse sampling [9, 34, 14]. Moreover, measured EFs are often based only on growing season emissions, which leads to a strong underestimation of emissions in cold regions [35, 11, 16]. EFs will be particularly underestimated from field measurements made at agricultural sites, where a significant proportion of N is removed through harvest or immobilization as described by fert_EF_red, thus leading to low EFs. These combined effects will lead to underestimation of deposition emissions when measured EFs are applied at non-agricultural sites in bottom up-N₂O emission frameworks.

10. **Lines 276-277: perhaps modify EF to "N₂O EF".**

This has been modified.

11. **Lines 310-314: A more recent reference: <https://doi.org/10.1016/j.cosust.2020.08.002> Also, as mentioned above, most studies do not include off-season emissions which can be high in temperate climates (<https://www.nature.com/articles/ngeo2907>).**

These references have been added.

12. **Figure 4: Considering that temperature is increasing at 2x the global rate in Canada I would have expected a temperature-dependence increase in N₂O emissions across Canada from 1940 to 2020.**

This is a good point. The temperature driven increase in emissions is proportional to both the baseline emissions, and the temperature increase. Although the temperature increase across most of Canada is large (captured in the CRUTEM dataset used to drive the temperature sensitivity calculation), the baseline emissions are low. As addressed in previous comments, sparse off-season sampling in cold regions may lead to an underestimation of emissions. We hope that this will improve as more data becomes available.

We realised in response to this point that a reference to the temperature anomaly dataset was missing. This has been added at L591:

Temperature anomalies from the CRUTEM-4.6.0 dataset were used to drive the temperature sensitivity calculation [22].

13. **Line 380: The description of the Python function could be moved to line 380 since this is when skipy is first mentioned.**

This has been corrected.

14. **Line 385: Was global mean WFPS estimated over time? Annually? There was only mention of annual mean precipitation above in line 369. Is annual WFPS used as another general aridity index? The seasonal temporal trend of WFPS over time is important for driving N₂O emissions. Also, the range of WFPS which impacts N₂O emissions is soil dependent since porosities differ**

(please see my comment below). WFPS should be defined in line 385 here where it is first mentioned.

WFPS was estimated annually using the CABLE model. Seasonal changes in WFPS are important to drive N₂O, however at this stage IsoTONE runs only at an annual time step, thus we aim to obtain an integrated annual picture of N inputs, processes and emissions. We hope to be able to run IsoTONE with sub-annual timesteps in the future. The point regarding porosity is addressed in response to comment 16.

15. Line 408: A tropospheric

This has been corrected.

16. Between lines 535 and 536: Why is the fraction of NH₃ always <0.05? I doubt this would be the case for some large agricultural regions when surface-applied urea is dominant. Regarding this statement: “Mean WFPS will not describe all variability in gas partitioning, given the highly variable and non-linear nature of N gas emissions, however, it provides the best available estimate based on the current status of experimental and modelling research.” Some models such as DNDC estimate soil oxygen content to drive N₂O emissions. Could you please explain how WFPS is applied as a driver for N₂O in your model since the impact should be soil specific with optimal N₂O production generally occurring a little above the upper water holding limit (field capacity)? This may be at 40% WFPS for a sandy loam but 85% for a Clay.

As stated, the fraction of NH₃ emitted is nearly always lower than 0.05 at an 0.5 degree gridcell; even large agricultural regions with surface-applied urea do not increase annual NH₃ emissions much above this proportion on this coarse grid scale. The CABLE model was used to estimate NH₃, and showed that the maximum fractional NH₃ emissions per grid cell were 5.6%, seen primarily in warm regions with intensive agriculture, such as parts of the western USA, India and Spain.

We believe from both our own results and from literature results that WFPS provides the best annual proxy for N gas emissions and N₂O emission pathways. Neither WFPS or soil oxygenation can account for the true microenvironment in which microbes conduct N cycling (see for example [18]). Soil texture will definitely also impact N cycling, substrate availability, and microsite chemistry. WFPS represents the amount of pore space filled with water and with air, thus describing the ability of gases and substrates to move through aqueous and gaseous environments, which is key in determining both substrate supply and soil oxygenation. We have clarified in the text:

Neither WFPS nor other potential proxies such as soil oxygenation can adequately describe the microenvironment in which microbes conduct N cycling [33, 18]. WFPS represents the amount of pore space filled with water and with air and can be compared between soils with different textures, thus it can be used as a proxy for the ability of gases and substrates to move through aqueous and gaseous environments, which is key in determining both substrate supply and soil oxygenation, and thus N cycling and gas production. Mean WFPS will not describe all variability in gas partitioning, given the highly variable and non-linear nature of N gas emissions, however it provides the best available estimate based on the current status of experimental and modelling research.

17. Lines 570 to 573 There may be publications which suggest EF is not strongly dependent on fertilisation rate, however, many other publications suggest that increases in N₂O emissions are non-linear as fertilizer rate increases and likewise there are lower EFs for well-managed systems. This is thought to be common knowledge amongst some scientists. Perhaps the impact of fertilizers on the global scale EFs is confounded by other drivers of N₂O which are not well characterized. Please see this policy brief from an OECD funded meeting which states “There is a clear non-linear relationship between fertilizer application and N₂O emissions, with emissions

increasing exponentially when application rates exceed plant N requirements.” [https://mopga.imk-ifu.kit.edu/sites/mopga/files/inline-files/Policy Brief - Nr-Workshop Garmisch final.pdf](https://mopga.imk-ifu.kit.edu/sites/mopga/files/inline-files/Policy%20Brief%20-%20Nr-Workshop%20Garmisch%20final.pdf). This has particularly been a problem in China where fertilizer has been greatly over applied (doi: 10.1016/j.eja.2021.126409). To be clear, I think it’s OK to use linear EFs in your global-scale study but it would be preferable to note that EFs may be strongly dependent on fertilizer rates and more research is needed to resolve the influence.

We find that there are some studies showing a strong non-linearity, and other studies which do not show this non-linearity (see the cited metaanalysis). We do agree with you that this is an open and important point and requires more research. Once reliable relationships between fertiliser excess and EF have been established, it would be a priority to incorporate this into IsoTONE. We have clarified this at L570-573:

EF changes with fertilisation rate were not considered: A recent metaanalysis showed that the dependence of EF on fertilisation rate is highly variable, and at the global scale EF is not strongly dependent on fertilisation rate [24], although some studies have shown a strong non-linear relationship between EF and N inputs at the site and regional scale [26]. [30] report that recent increases in EF for direct soil emissions are likely due to climate change feedbacks and interactions, rather than a direct increase in EF due to increasing fertiliser application. As more evidence regarding the non-linearity of fertiliser EFs becomes available, this will be incorporated into the IsoTONE model.

18. **Line 668 “show” should be “shown”**

This has been corrected.

19. **Line 817-818: Was the impact of N availability low due to the highly variable soil N since it “showed no correlation between the point and gridded datasets” as indicated from lines 716-717? The relationship between EF and N availability is important, however, to capture the relationship a robust measurement or process-based modelling study is needed. I would suggest that the authors be careful not to discount this relationship as many research studies and also agricultural producer groups stress the importance of N use efficiency in cropping systems on N₂O emissions.**

The lack of a strong relationship may be due to sub-grid cell processes and strong heterogeneity; as you state, this is best captured and the site scale using combined measurements and process-based modelling. The disconnect between point and gridded datasets is furthermore driven by sub-gridcell heterogeneity. We have rephrased to emphasise that our findings do not discount the importance of this relationship:

The lack of a strong relationship between soil N inputs and availability and EF may also be due to sub-grid cell processes and strong heterogeneity in soil nitrogen, which cannot be captured on the coarse spatial scale currently used in the IsoTONE framework. Our results are consistent with numerous studies, which show that although more N availability clearly leads to more N gas production, the relationship between N availability and N gas EFs is unclear [9, 20, 29, 15].

20. **Figure A.5 Likely we can expect a smaller N₂O/(N₂O + N₂) ratio in fine-textured soils (denitrification is more apt to go to N₂ in clay soils). Have the authors considered developing equations with coarse, medium and fine-textured soils? Again, as mentioned above, WFPS is not a very good measure of N gas emissions across soil types.**

Once enough data is available to consider separate parameterisations in different soil types, or to directly include soil texture as an input to the parameterisation, this would be a very high priority to investigate and add to the model. As shown in the figure, data is very sparse, so this is not currently

possible. WFPS represents the best proxy across different soil types based on currently available data, as discussed under point 10.

21. **Figure A.10 Please revise the caption as b) map of N lost to nitrification is not mentioned. Also, why does the proportion of N₂O from both denitrification and nitrification decline over time? What is the other source of N₂O emissions?**

This has been addressed under comment 13 from Reviewer 1.

Other minor changes

1. Several of the author affiliations were corrected and updated.
2. All instances of 'in situ' were corrected to *in situ*
3. Figure 4: The colour scale for Panel *b* was changed to be identical to Panel *a*.
4. Figure 6: "¹⁵N measurements" were put in a red box, as these are observational data.
5. Figure A11: ':' was removed from y axis titles to avoid any confusion that this is a ratio.
6. Numbering of figures, tables and subsections in the methods and supplementary materials were edited in line with the formatting guidelines of Nature Communications.

References

- [1] M. Barthel, M. Bauters, S. Baumgartner, T. W. Drake, N. M. Bey, G. Bush, P. Boeckx, C. I. Botefa, N. Dériaz, G. L. Ekamba, N. Gallarotti, F. M. Mbayu, J. K. Mugula, I. A. Makelele, C. E. Mbongo, J. Mohn, J. Z. Manda, D. M. Mpambi, L. C. Ntaboba, M. B. Rukeza, R. G. Spencer, L. Summerauer, B. Vanlauwe, K. Van Oost, B. Wolf, and J. Six. Low N₂O and variable CH₄ fluxes from tropical forest soils of the Congo Basin. *Nature Communications*, 13(1):1–8, 2022.
- [2] S. Baumgartner, M. Bauters, M. Barthel, T. W. Drake, L. C. Ntaboba, B. M. Bazirake, J. Six, P. Boeckx, and K. Van Oost. Stable isotope signatures of soil nitrogen on an environmental-geomorphic gradient within the Congo Basin. *Soil*, 7(1):83–94, 2021.
- [3] M. Bauters, H. Verbeeck, M. Demol, S. Bruneel, C. Taveirne, D. Van Der Heyden, L. Cizungu, and P. Boeckx. Parallel functional and stoichiometric trait shifts in South American and African forest communities with elevation. *Biogeosciences*, 14(23):5313–5321, 2017.
- [4] M. Bauters, H. Verbeeck, S. Doetterl, E. Ampoorter, G. Baert, P. Vermeir, K. Verheyen, and P. Boeckx. Functional Composition of Tree Communities Changed Topsoil Properties in an Old Experimental Tropical Plantation. *Ecosystems*, 20(5):861–871, 2017.
- [5] M. Bauters, H. Verbeeck, T. Rütting, M. Barthel, B. Bazirake Mujinya, F. Bamba, S. Bodé, F. Boyemba, E. Bulonza, E. Carlsson, L. Eriksson, I. Makelele, J. Six, L. Cizungu Ntaboba, and P. Boeckx. Contrasting nitrogen fluxes in African tropical forests of the Congo Basin. *Ecological Monographs*, 89(1), 2019.
- [6] M. Bauters, O. Verceleyen, B. Vanlauwe, J. Six, B. Bonyoma, H. Badjoko, W. Hubau, A. Hoyt, M. Boudin, H. Verbeeck, and P. Boeckx. Long-term recovery of the functional community assembly and carbon pools in an African tropical forest succession. *Biotropica*, 51(3):319–329, 2019.
- [7] A. Bissett, A. Fitzgerald, T. Meintjes, P. Mele, F. Reith, P. Dennis, M. Breed, B. Brown, M. Brown, J. Brugger, M. Byrne, S. Caddy-Retalic, B. Carmody, D. Coates, C. Correa, B. Ferrari, V. Gupta, K. Hamonts, A. Haslem, P. Hugenholtz, M. Karan, J. Koval, A. Lowe, S. Macdonald, L. McGrath, D. Martin, M. Morgan, K. North, C. Paungfoo-Lonhienne, E. Pendall, L. Phillips, R. Pirzl, J. Powell, M. Ragan, S. Schmidt, N. Seymour, I. Snape, J. Stephen, M. Stevens, M. Tinning, K. Williams, Y. Yeoh, C. Zammit, and A. Young. Introducing BASE: the Biomes of Australian Soil Environments soil microbial diversity database. *GigaScience*, 5, 2016.
- [8] D. Brenner, R. Amundson, T. Baisden, C. Kendall, and J. Harden. N variation with time in a California annual grassland ecosystem. *Geochimica et Cosmochimica Acta*, 65(22):4171–4186, 2001.
- [9] K. Butterbach-Bahl, E. M. Baggs, M. Dannenmann, R. Kiese, and S. Zechmeister-Boltenstern. Nitrous oxide emissions from soils: how well do we understand the processes and their controls? *Philosophical transactions of the Royal Society of London, Series B, Biological sciences*, 368(1621), 2013.
- [10] K. Butterbach-Bahl, F. Stange, H. Papen, and C. Li. Regional inventory of nitric oxide and nitrous oxide emissions for forest soils of southeast Germany using the biogeochemical model PnET-N-DNDC. *Journal of Geophysical Research - Atmospheres*, 106(D24):34155–34166, 2001.
- [11] E. Byers, M. A. Bleken, and P. Dörsch. Winter N₂O accumulation and emission in sub-boreal grassland soil depend on clover proportion and soil ph. *Environmental Research Communications*, 3(1), 2021.

- [12] C. Chen, Y. Jia, Y. Chen, I. Mehmood, Y. Fang, and G. Wang. Nitrogen isotopic composition of plants and soil in an arid mountainous terrain: South slope versus north slope. *Biogeosciences*, 15(1):369–377, 2018.
- [13] T. J. Clough, G. J. Lanigan, C. A. De Klein, M. S. Samad, S. E. Morales, D. Rex, L. R. Bakken, C. Johns, L. M. Condron, J. Grant, and K. G. Richards. Influence of soil moisture on codenitrification fluxes from a urea-affected pasture soil. *Scientific Reports*, 7(1):1–12, 2017.
- [14] K. A. Congreves, C. Wagner-Riddle, B. C. Si, and T. J. Clough. Nitrous oxide emissions and biogeochemical responses to soil freezing-thawing and drying-wetting. *Soil Biology and Biochemistry*, 117(October 2017):5–15, 2018.
- [15] X. Cui, F. Zhou, P. Ciais, E. A. Davidson, F. N. Tubiello, X. Niu, X. Ju, J. G. Canadell, A. F. Bouwman, R. Jackson, N. Mueller, X. Zheng, D. Kanter, H. Tian, W. Adalibieke, Y. Bo, Q. Wang, X. Zhan, and D. Zhu. Global mapping of crop-specific emission factors highlights hotspots of nitrous oxide mitigation. *Nature Food*, In press, 2021.
- [16] P. Doersch, I. Sturite, and S. Trier Kjaer. High off-season nitrous oxide emissions negate potential soil C-gain from cover crops in boreal cereal cropping (EGU22-3066). *EGU General Assembly 2022*, 2022.
- [17] N. Gallarotti, M. Barthel, E. Verhoeven, P. P. Engil Isadora, M. Bauters, S. Baumgartner, T. W. Drake, P. Boeckx, J. Mohn, M. Longepierre, J. Kalume Mugula, I. Ahanamungu Makelele, L. Cizungu Ntaboba, and J. Six. In-depth analysis of N₂O fluxes in tropical forest soils of the Congo Basin combining isotope and functional gene analysis. *The ISME Journal*, 2021.
- [18] E. Harris, E. Diaz-Pines, E. Stoll, M. Schlöter, S. Schulz, C. Duffner, K. Li, K. Moore, J. Ingrisch, D. Reinthaler, S. Zechmeister-Boltenstern, S. Glatzel, and M. Bahn. Denitrifying pathways dominate nitrous oxide emissions from managed grassland during drought and rewetting. *Science Advances*, 7(6):eabb7118, 2021.
- [19] E. Harris, S. Henne, C. Hüglin, C. Zellweger, B. Tuzson, E. Ibraim, L. Emmenegger, and J. Mohn. Tracking nitrous oxide emission processes at a suburban site with semicontinuous, in situ measurements of isotopic composition. *Journal of Geophysical Research - Atmospheres*, 122:1–21, 2017.
- [20] M. Inatomi, T. Hajima, and A. Ito. Fraction of nitrous oxide production in nitrification and its effect on total soil emission: A meta-analysis and global-scale sensitivity analysis using a process-based model. *Plos One*, 14(7):e0219159, 2019.
- [21] P. W. Inglett, K. R. Reddy, S. Newman, and B. Lorenzen. Increased soil stable nitrogen isotopic ratio following phosphorus enrichment: Historical patterns and tests of two hypotheses in a phosphorus-limited wetland. *Oecologia*, 153(1):99–109, 2007.
- [22] P. Jones, D. Lister, T. Osborn, C. Harpham, M. Salmon, and C. Morice. Hemispheric and large-scale land surface air temperature variations: An extensive revision and an update to 2010. *Journal of Geophysical Research*, 117:D05127, 2012.
- [23] M. Kesik, P. Ambus, R. Baritz, N. Brüeggemann, K. Butterbach-Bahl, M. Damm, J. Duyzer, L. Horvath, R. Kiese, B. Kitzler, A. Leip, C. Li, M. Pihlatie, K. Pilegaard, G. Seufert, D. Simpson, U. Skiba, G. Smiatek, T. Vesala, and S. Zechmeister-Boltenstern. Inventories of N₂O and NO emissions from European forest soils. *Biogeosciences*, 2:353–375, 2005.
- [24] D. G. Kim, G. Hernandez-Ramirez, and D. Giltrap. Linear and nonlinear dependency of direct nitrous oxide emissions on fertilizer nitrogen input: A meta-analysis. *Agriculture, Ecosystems and Environment*, 168:53–65, 2013.

- [25] M. Prokopiou, P. Martinerie, C. J. Sapart, E. Witrant, G. Monteil, K. Ishijima, S. Bernard, J. Kaiser, I. Levin, T. Blunier, D. Etheridge, E. Dlugokencky, R. S. Van De Wal, and T. Röckmann. Constraining N₂O emissions since 1940 using firn air isotope measurements in both hemispheres. *Atmospheric Chemistry and Physics*, 17(7):4539–4564, 2017.
- [26] C. Scheer, D. Pelster, K. Butterbach-Bahl, O. Cleemput, D. van Kanter, W. Winiwarter, S. Ogle, P. Boeckx, K. Fuchs, E. Baggs, L. Bakken, L. Barton, L. Cardenas, T. Clough, S. DelGrosso, C. Dorich, J. Friedl, C. Hu, S. Leitner, R. Massad, S. Peterson, U. Skiba, W. Smith, G. Subbarao, and I. Vogeler. Addressing nitrous oxide: An often ignored climate and ozone threat. Technical report, 2019.
- [27] W. Smith, B. Grant, Z. Qi, W. He, B. Qian, Q. Jing, A. VanderZaag, C. F. Drury, M. St. Luce, and C. Wagner-Riddle. Towards an improved methodology for modelling climate change impacts on cropping systems in cool climates. *Science of the Total Environment*, 728:138845, 2020.
- [28] R. Thompson. Documentation of N₂O flux service: Description of the N₂O inversion production chain. Technical report, Copernicus Atmospheric Monitoring Service, 2021.
- [29] R. L. Thompson, L. Lassaletta, P. K. Patra, C. Wilson, K. C. Wells, A. Gressent, E. N. Koffi, M. P. Chipperfield, W. Winiwarter, E. A. Davidson, H. Tian, and J. Canadell. Acceleration of global N₂O emissions seen from two decades of atmospheric inversion. *Nature Climate Change*, page 8, 2019.
- [30] H. Tian, R. Xu, J. G. Canadell, R. L. Thompson, W. Winiwarter, P. Suntharalingam, E. A. Davidson, P. Ciais, R. B. Jackson, G. Janssens-maenhout, M. J. Prather, P. Regnier, N. Pan, S. Pan, G. P. Peters, H. Shi, F. N. Tubiello, S. Zaehle, F. Zhou, A. Arneth, G. Battaglia, S. Berthet, L. Bopp, A. F. Bouwman, E. T. Buitenhuis, J. Chang, M. P. Chipperfield, S. R. S. Dangal, E. Dlugokencky, J. W. Elkins, B. D. Eyre, B. Fu, B. Hall, A. Ito, F. Joos, P. B. Krummel, A. Landolfi, G. G. Laruelle, R. Lauerwald, W. Li, S. Lienert, T. Maavara, M. Macleod, D. B. Millet, S. Olin, P. K. Patra, R. G. Prinn, P. A. Raymond, D. J. Ruiz, G. R. Werf, N. Vuichard, J. Wang, R. F. Weiss, K. C. Wells, C. Wilson, J. Yang, and Y. Yao. A comprehensive quantification of global nitrous oxide sources and sinks. *Nature*, 586, 2020.
- [31] H. Tian, J. Yang, R. Xu, C. Lu, J. G. Canadell, E. A. Davidson, R. B. Jackson, A. Arneth, J. Chang, P. Ciais, S. Gerber, A. Ito, F. Joos, S. Lienert, P. Messina, S. Olin, S. Pan, C. Peng, E. Saikawa, R. L. Thompson, N. Vuichard, W. Winiwarter, S. Zaehle, and B. Zhang. Global soil nitrous oxide emissions since the preindustrial era estimated by an ensemble of terrestrial biosphere models: Magnitude, attribution, and uncertainty. *Global Change Biology*, 25(2):640–659, 2019.
- [32] S. Toyoda, N. Kuroki, N. Yoshida, K. Ishijima, Y. Tohjima, and T. Machida. Decadal time series of tropospheric abundance of N₂O isotopomers and isotopologues in the northern hemisphere obtained by the long-term observation at Hateruma Island, Japan. *Journal of Geophysical Research - Atmospheres*, 118:1–13, 2013.
- [33] H. Vereecken, A. Schnepf, J. Hopmans, M. Javaux, D. Or, T. Roose, J. Vanderborght, M. Young, W. Amelung, M. Aitkenhead, S. Allison, S. Assouline, P. Baveye, M. Berli, N. Brüggemann, P. Finke, M. Flury, T. Gaiser, G. Govers, T. Ghezzehei, P. Hallett, and K. Lamorski. Modeling Soil Processes: Review, Key challenges and New Perspectives. *Vadose Zone Journal*, 15(5):1–57, 2016.
- [34] C. Wagner-Riddle, E. M. Baggs, T. J. Clough, K. Fuchs, and S. O. Petersen. Mitigation of nitrous oxide emissions in the context of nitrogen loss reduction from agroecosystems: managing hot spots and hot moments. *Current Opinion in Environmental Sustainability*, 47:46–53, 2020.
- [35] C. Wagner-Riddle, K. A. Congreves, D. Abalos, A. A. Berg, S. E. Brown, J. T. Ambadan, X. Gao, and M. Tenuta. Globally important nitrous oxide emissions from croplands induced by freeze-thaw cycles. *Nature Geoscience*, 10(4):279–283, 2017.

- [36] J. Wei, W. Amelung, E. Lehdorff, M. Schloter, H. Vereecken, and N. Brüggemann. N₂O and NO_x emissions by reactions of nitrite with soil organic matter of a Norway spruce forest. *Biogeochemistry*, 132(3):325–342, 2017.
- [37] Y. Xu, J. He, W. Cheng, X. Xing, and L. Li. Natural ¹⁵N abundance in soils and plants in relation to N cycling in a rangeland in Inner Mongolia. *Journal of Plant Ecology*, 3(3):201–207, 2010.
- [38] L. Yu, E. Harris, S. Henne, S. Eggleston, M. Steinbacher, L. Emmenegger, C. Zellweger, and J. Mohn. Atmospheric nitrous oxide isotopes observed at the high-altitude research station Jungfrauoch, Switzerland. *Atmospheric Chemistry and Physics*, 20:6495–6519, 2020.
- [39] L. Yu, E. Harris, D. Lewicka-Szczebak, and J. Mohn. What can we learn from N₂O isotope data? Analytics, processes and modelling. *Rapid Communications in Mass Spectrometry*, 2020.

REVIEWERS' COMMENTS

Reviewer #4 (Remarks to the Author):

The authors responded very well to my concerns and updated the manuscript appropriately. I have no further comments and suggest the paper proceed to publication.